# Channel Adapter for Time Series Foundation Models in Zero-shot Multivariate Forecasting

**Dongyuan Li** [1]  **Renhe Jiang** [†1]  **Shun Zheng** [2]  **Zheng Dong** [1]  **Haotian Gao** [1]  **Ying Zhang** [3]  **Jiang Bian** [2]

## Abstract

Time Series Foundation Models (TSFMs) have achieved strong performance in univariate time series forecasting. However, most TSFMs rely on channel-independent pre-training that models each variable separately, limiting their ability to leverage inter-channel information that is crucial in real-world multivariate systems. Motivated by this limitation, we propose **ChaTSFM**, a lightweight plug-and-play channel adapter that allows frozen TSFMs to leverage multivariate correlations in a zero-shot setting. ChaTSFM first builds a budgeted pre-training dataset to cover diverse heterogeneous inter-channel dependency patterns. It then uses data-derived domain descriptors to learn a dataset-conditioned inter-channel similarity measure that reduces cross-domain metric distortion. Finally, it injects sparse inter-channel information via gated refinement, leveraging multivariate information without degrading intra-channel temporal dynamics. Extensive experiments on nine benchmarks validate the effectiveness of ChaTSFM, demonstrating consistent zero-shot improvements over four best-performing TSFMs while maintaining scalable deployment. Code is available at https://github.com/Clearloveyuan/ChaTSFM.

## 1. Introduction

Multivariate time series forecasting (MTSF), a fundamental task in time series analysis, has emerged as a key focus in AI due to its broad applications in transportation, finance, and environment (Qiu et al., 2024a; Liang et al., 2024; Li et al., 2026b; Liu et al., 2026). Fueled by data scaling and

general intelligence, the field has shifted from early architectural adaptations (Nie et al., 2023; Chen et al., 2024a) toward Time Series Foundation Models (Meyer et al., 2025; Ansari et al., 2025). By capturing transferable intra-channel patterns, TSFMs deliver substantial gains. In particular, GridStor leveraged TimeGPT (Garza et al., 2023) for electricity price forecasting, achieving a 15% accuracy boost alongside a 50% reduction in cloud costs.

TSFMs predominantly adopt the Channel-Independent (CI) paradigm to ensure scalability across diverse datasets. However, by modeling each time series in isolation, CI inherently overlooks critical inter-channel dependencies (Liu et al., 2025b; Wang et al., 2025; Xiao et al., 2025). To recover these spatial relationships, existing methods mainly take two directions: (1) Parameter adaptation, ranging from full fine-tuning (Ekambaram et al., 2024; Lee et al., 2024) to parameter efficient fine-tuning (Beichter et al., 2025; Benechehab et al., 2025); and (ii) Input augmentation, which enriches the context with retrieved exemplars (Han et al., 2025b) or exogenous covariates (Qin et al., 2025; Han et al., 2025a). However, these approaches often require dataset-specific parameter updates or costly retrieval, increasing computational overhead and potentially compromising the zero-shot generalization of TSFMs. This dilemma motivates our primary research question: *How can we design a lightweight, plug-and-play module to empower TSFMs for better zero-shot multivariate time series forecasting?*

To answer this question, we identify three fundamental challenges: **(1) Structural Redundancy.** A zero-shot plugin demands a pre-training corpus that encompasses a wide spectrum of inter-channel dependencies. However, naively aggregating all available datasets introduces redundancy and computational burden, while random sampling risks missing rare but critical structural patterns. Therefore, we must strategically curate a compact and informative corpus that maximizes geometric diversity under a strict computational budget. **(2) Geometric Distortion.** Inter-channel dependencies exhibit significant domain variance, ranging from volatile financial correlations to stable physical couplings. Consequently, a fixed, universal distance metric fails to capture these nuances, leading to geometric distortion where identical embedding distances signify disparate dependency

---

† Corresponding Author. [1]The University of Tokyo [2]Microsoft Research Asia [3]RIKEN AIP. Correspondence to: Renhe Jiang <Jiangrh@csis.u-tokyo.ac.jp>.

*Proceedings of the 43rd International Conference on Machine Learning*, Seoul, South Korea. PMLR 306, 2026. Copyright 2026 by the author(s).

*Table 1.* Comparison with state-of-the-art TSFM plugins and adapters, including STAR (Cheng et al., 2025), UniCA (Han et al., 2025a), Cora (Qin et al., 2025), ICF (Faw et al., 2025), ELF (Lee et al., 2025), ICM (Żukowska et al., 2024), TTM (Ekambaram et al., 2024), AdaPTS (Benechehab et al., 2025), PCD (Lee et al., 2024), PEFT (Beichter et al., 2025), Gen-P-Tuning (Liu et al., 2024b).

| Capabilities | STAR | UniCA | CoRA | ICF | ELF | ICM | TTM | AdaPTS | PCD | PEFT | Gen-P-Tuning | ChaTSFM(Ours) |
|---|---|---|---|---|---|---|---|---|---|---|---|---|
| Multivariate Supportive | ✓ | ✗ | ✗ | ✓ | ✓ | ✓ | ✓ | ✓ | ✓ | ✓ | ✓ | ✓ |
| Forecasting Supportive | ✗ | ✓ | ✓ | ✓ | ✓ | ✓ | ✓ | ✓ | ✓ | ✓ | ✓ | ✓ |
| Variable-Channel Zero-shot | ✗ | ✗ | ✗ | ✓ | ✗ | ✓ | ✗ | ✗ | ✗ | ✗ | ✗ | ✓ |
| Domain-aware Zero-shot | ✗ | ✗ | ✗ | ✓ | ✓ | ✗ | ✗ | ✗ | ✓ | ✗ | ✗ | ✓ |
| Architecture-agnostic Plugin | ✓ | ✓ | ✓ | ✗ | ✗ | ✗ | ✗ | ✓ | ✗ | ✗ | ✗ | ✓ |

strengths. Overcoming this requires dynamically calibrating the metric space conditioned on domain characteristics, enabling accurate affinity estimation without domain-specific supervision. **(3) Spurious Correlations**. In zero-shot settings with unknown causal graphs, blindly introducing multivariate interactions risks injecting noise from irrelevant channels, which can overwhelm the target variable. Crucially, the plugin must achieve a channel-adapter design: dynamically inducing a sparse, high-fidelity topology that captures critical interactions while filtering out noise.

To tackle these challenges, we propose **ChaTSFM**, a lightweight yet theoretically grounded framework to empower TSFMs with zero-shot channel-adapter capabilities. **First**, to resolve structural redundancy, we move beyond naive data sampling by representing each candidate domain as a Gaussian proxy on a statistical manifold, subsequently employing a greedy selection strategy to curate a compact pre-training corpus that maximizes Wasserstein-based geometric diversity and enables the plugin to learn a universal prior of channel interactions under a minimal data budget. **Second**, to mitigate geometric distortion, we introduce a Hyper-Network that dynamically generates domain-specific metric parameters from input statistics, thereby calibrating the similarity space on-the-fly to ensure accurate dependency estimation across disparate physical domains without task-specific supervision. **Finally**, to eliminate spurious correlations, we implement a sparse, high-fidelity interaction mechanism that induces a channel-adapter topology with linear scalability via nearest neighbor search. A gated residual design then provides a selective inductive bias, refining predictions only when inter-channel context yields information gain. Main contributions are summarized as:

- We propose ChaTSFM, a lightweight framework that pioneers zero-shot, channel-partial capabilities for TSFMs. By decoupling temporal dynamics from spatial dependencies, ChaTSFM provides multivariate gains across diverse time series foundation models without dataset-specific fine-tuning.

- We recognize that channel dependencies vary significantly across domains and introduce geometric metric adaptation to intrinsically characterize these manifold-aware relationships, coupled with a selective channel adapter that filters noise to distill high-fidelity signals.

- Extensive experiments on nine datasets show that ChaTSFM significantly boosts zero-shot performance with high computational efficiency, often even surpassing full-shot fine-tuned baselines. Its architecture-agnostic design further ensures seamless integration across various TSFM backbones.

## 2. Related Work

**Multivariate Time Series Forecasting** is to predict future values of multiple variables based on their historical observations. To capture both intra-channel temporal dynamics and inter-channel spatial dependencies, many architectures are proposed, such as **Statistical Model** (Zhang, 2003), **RNNs** (Lin et al., 2023), **CNNs** (Yi et al., 2023b; Wu et al., 2023), **MLPs** (Das et al., 2023; Zeng et al., 2023a; Hu et al., 2025a), **Transformers** (Li et al., 2026a; Liu et al., 2022; Nie et al., 2023; Zhang & Yan, 2023; Liu et al., 2024c; Feng et al., 2024a; Liu et al., 2025a; Hu et al., 2025b), and **GNNs** (Chen et al., 2023; 2024b; Qiu et al., 2024b; Yi et al., 2023a; Cai et al., 2024; Shang et al., 2024). Existing spatiotemporal forecasting models (Liu et al., 2023; Jiang et al., 2023; Gao et al., 2024; Dong et al., 2024; Gao et al., 2026) often combine multiple architectural components to preserve channel-specific temporal dynamics while modeling cross-channel dependencies.

**Time Series Foundation Models** could be broadly grouped into **Decoder-only** models like TimesFM and Sundial (Das et al., 2024; Liu et al., 2024d; Rasul et al., 2024; Jin et al., 2024; Shi et al., 2024), **Encoder-only** models like Moirai 2.0 and Moment (Woo et al., 2024; Goswami et al., 2024; Feofanov et al., 2025; Feng et al., 2024b), **Encoder-Decoder** models like Chronos and TimeGPT (Ansari et al., 2024; Garza et al., 2023; Cao et al.), and **MLP-based** models like TTMs and TSMixer (Ekambaram et al., 2024; Wang et al., 2024). Current TSFMs predominantly operate under the CI paradigm to ensure distribution stability during pre-training. While robust, CI inherently overlooks the critical cross-channel synergies required for complex multivariate forecasting. To bridge this gap, existing efforts primarily follow two trajectories: First, they adapt TSFMs via parameter updates, ranging from full fine-tuning (Ekambaram et al., 2024; Benechehab et al., 2025) to lightweight in-context (Faw et al., 2025), LoRA (Beichter et al., 2025)

and P-tuning (Liu et al., 2024b). Second, they improve TSFMs without substantial parameter changes by augmenting the input, such as retrieval-augmented inputs (Liu et al., 2024a; Han et al., 2025b), LLM-aligned multimodal signals (Zhong et al., 2025; Pan et al., 2024), or exogenous covariates (Qin et al., 2025; Han et al., 2025a). As shown in Table 1, in contrast to existing paradigms, ChaTSFM shifts the focus from parameter adaptation to zero-shot topology inference. While conventional methods typically rely on task-specific recalibration that may undermine the stability of foundation priors, ChaTSFM treats cross-channel modeling as a dynamic alignment problem. By decoupling spatial dependencies from pre-trained temporal features, our approach eliminates the need for data-intensive fine-tuning and avoids the inherent noise of static interaction structures. This results in an architecture-agnostic enhancement that yields stable multivariate gains while strictly preserving the zero-shot integrity and computational efficiency of TSFMs.

## 3. Preliminary

Let $\boldsymbol{X} = [\boldsymbol{x}_1, \ldots, \boldsymbol{x}_C] \in \mathbb{R}^{C \times T}$ denote a multivariate time series, where $C$ is the number of channels and $T$ is the lookback horizon. MTSF aims to build a predictive model $f_{\text{model}}$: $\mathbb{R}^{C \times T} \to \mathbb{R}^{C \times T'}$ that takes historical observations $\boldsymbol{X}$ as input and generates future predictions $\widehat{\boldsymbol{X}} \in \mathbb{R}^{C \times T'}$, where $T'$ is the forecasting horizon. Here, rather than building $f_{\text{model}}$, we develop a plugin $f_{\text{plugin}}$: $\mathbb{R}^{C \times T'} \to \mathbb{R}^{C \times T'}$ that post-processes the predictions from existing TSFMs, *i.e.,* given model's output $\widehat{\boldsymbol{X}}^{\text{model}} = f_{\text{model}}(\boldsymbol{X})$, our plugin refines predictions to obtain improved forecasts $\widehat{\boldsymbol{X}} = f_{\text{plugin}}(\widehat{\boldsymbol{X}}^{\text{model}})$.

## 4. Method

**Data Flow Overview.** ChaTSFM follows a "select–adapt–refine" data flow to empower TSFMs with zero-shot forecasting (Figure 1). It operates in two phases. In the offline corpus-curation phase (Sec. 4.1), Generative Domain Selection selects a geometrically diverse pre-training corpus, providing broad domain coverage for learning the adaptation function. In the online target-inference phase (Secs. 4.2–4.3), Geometric Metric Adaptation takes the statistical features $\phi(X)$ of a new target dataset as input and generates domain-adaptive metric parameters on the fly, thereby correcting inter-domain distortions. Finally, Channel-Partial Modeling refines the TSFM outputs through a gated residual mechanism that captures inter-channel synergies while preserving the pre-trained temporal priors.

### 4.1. Generative Domain Selection

The success of TSFMs hinges critically on the diversity and quality of pre-training data (Woo et al., 2024; Liu et al., 2024d). However, to build a lightweight plugin, we must

maximize the coverage of diverse multivariate interaction patterns under a strictly limited data budget. Inspired by domain-aware representation learning (Wang et al.; Yuan et al., 2025), we argue that decoupling temporal dynamics from spatial structures is key to reducing redundancy. A naive alternative is to compare domains directly in the raw series space, yet such a comparison is dominated by superficial factors (e.g., scale, length, and sampling rate) that obscure the cross-channel dependency structure we actually aim to cover. This motivates a *generative* rather than *descriptive* view: we characterize a domain not by the surface statistics of its observations, but by the latent dynamical law that produces them. A stochastic differential equation (SDE) is particularly suited to this end, as it offers an explicit decoupling between a *drift* term governing deterministic temporal evolution and a *diffusion* term governing stochastic cross-variable interactions—precisely the two degrees of freedom we wish to separate. Following this rationale (Zeng et al., 2023b; Oh et al., 2024), we therefore view a fine-grained time-series domain as a stochastic process:

**Definition 4.1.** Consider a multivariate time series $\mathbf{X} \in \mathbb{R}^{C \times T}$ generated by a latent continuous-time system. A fine-grained domain $\mathcal{D}$ is defined by the generative parameter tuple $(\boldsymbol{\Theta}_{\text{temp}}, \boldsymbol{\Theta}_{\text{spat}})$ of the following stochastic differential equation: $d\mathbf{x}_t = \mathbf{f}(\mathbf{x}_t; \boldsymbol{\Theta}_{\text{temp}}) \, dt + \mathbf{g}(\mathbf{x}_t; \boldsymbol{\Theta}_{\text{spat}}) \, d\mathbf{W}_t$, where $\mathbf{f}(\cdot)$ is the drift vector field controlling *deterministic temporal evolution*, $\mathbf{g}(\cdot)$ is the diffusion term capturing *stochastic cross-variable interactions*, and $\mathbf{W}_t$ denotes a standard Wiener process.

Under Definition 4.1, the domain $\mathcal{D}$ induces a (path-level) distribution over observed series $\mathbf{X}$ via the latent diffusion dynamics. However, directly modeling or comparing such raw series-level distributions is unstable and computationally expensive at the dataset scale. Instead, we summarize each sample path by a $k$-dimensional statistical feature map $\phi : \mathbb{R}^{C \times T} \to \mathbb{R}^k$, which induces a feature-space distribution $p(\phi(\mathbf{X})|\mathcal{D})$. We then use its low-order moments, i.e., the Gaussian parameters $(\boldsymbol{\mu}_{\mathcal{D}}, \boldsymbol{\Sigma}_{\mathcal{D}})$, as a lightweight and comparable representation of the domain, formalized below.

**Proposition 4.2** (Gaussian-based Domain Representation). *Assume Definition 4.1 is stationary and admits a stationary distribution $p_{\mathcal{D}}$. Let $\phi : \mathbb{R}^{C \times T} \to \mathbb{R}^k$ be the extracted domain statistics and define $\boldsymbol{\mu}_{\mathcal{D}} = \mathbb{E}[\phi(\mathbf{X})|\mathcal{D}]$ and $\boldsymbol{\Sigma}_{\mathcal{D}} = \text{Cov}[\phi(\mathbf{X})|\mathcal{D}]$. Then $(\boldsymbol{\mu}_{\mathcal{D}}, \boldsymbol{\Sigma}_{\mathcal{D}})$ provides a Gaussian surrogate of $\mathcal{D}$ with $\boldsymbol{\mu}_{\mathcal{D}} \propto \mathbb{E}_{x \sim p_{\mathcal{D}}}[\mathbf{f}(x; \boldsymbol{\Theta}_{temp})], \boldsymbol{\Sigma}_{\mathcal{D}} \propto \mathbb{E}_{x \sim p_{\mathcal{D}}}[\mathbf{g}(x; \boldsymbol{\Theta}_{spat})\mathbf{g}(x; \boldsymbol{\Theta}_{spat})^\top]$, up to an affine transformation of $\phi(\mathbf{X})$ (e.g., feature normalization).*

Guided by Proposition 4.2, we represent each domain $\mathcal{D}$ by a Gaussian proxy $\mathcal{N}(\boldsymbol{\mu}_{\mathcal{D}}, \boldsymbol{\Sigma}_{\mathcal{D}})$ estimated from $k{=}7$ statistical features $\phi(\mathbf{X})$ over $M{=}1{,}000$ sampled instances (Appendix E). Here, $\boldsymbol{\mu}_{\mathcal{D}}$ and $\boldsymbol{\Sigma}_{\mathcal{D}}$ act as macroscopic surrogates of the latent SDE, capturing deterministic tempo-

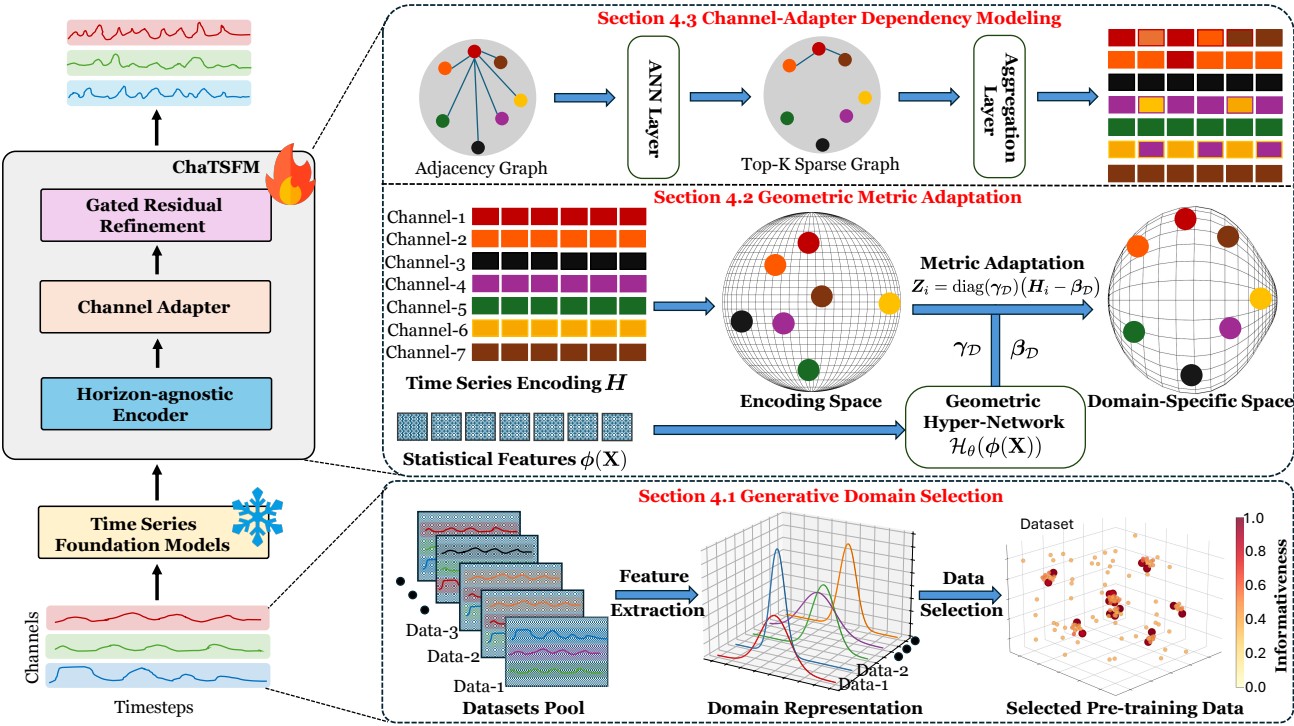

*Figure 1.* The overall framework of ChaTSFM, featuring a three-stage pipeline: (i) generative domain selection for data curation, (ii) geometric metric adaptation for manifold rectification, and (iii) channel-adapter dependency modeling for topology induction.

ral tendencies (drift) and stochastic cross-variable interactions (diffusion), respectively. Given a candidate pool $\mathcal{U} = \{\mathcal{D}_1, \ldots, \mathcal{D}_N\}$, we curate a compact subset $\mathcal{M}$ of size $K$ ($K \ll |\mathcal{U}|$) via a greedy rule that balances informativeness and generative diversity. At each step, we add

$$\mathcal{D}^* = \underset{\mathcal{D} \in \mathcal{U} \setminus \mathcal{M}}{\arg\max}(\underbrace{\lambda \log \det(\mathbf{\Sigma}_{\mathcal{D}} + \epsilon \mathbf{I})}_{\text{Part I: Informativeness}} + \underbrace{\min_{\mathcal{D}_j \in \mathcal{M}} W_2(\mathcal{D}, \mathcal{D}_j)}_{\text{Part II: Diversity}}),$$

(1)

and update $\mathcal{M} \leftarrow \mathcal{M} \cup \{\mathcal{D}^*\}$ until $|\mathcal{M}| = K$. Part I measures feature-space uncertainty via the log-determinant of $\mathbf{\Sigma}_{\mathcal{D}}$, favoring domains with richer statistical variability. Part II encourages coverage by maximizing the Gaussian $W_2$ distance to $\mathcal{S}$, which admits a drift-diffusion decomposition:

$$W_2^2(\mathcal{D}, \mathcal{D}_j) = \underbrace{\|\boldsymbol{\mu} - \boldsymbol{\mu}_j\|_2^2}_{\text{Drift Diversity}} + \underbrace{\text{Tr}(\mathbf{\Sigma} + \mathbf{\Sigma}_j - 2(\mathbf{\Sigma}^{\frac{1}{2}} \mathbf{\Sigma}_j \mathbf{\Sigma}^{\frac{1}{2}})^{\frac{1}{2}})}_{\text{Diffusion Diversity}},$$

(2)

where the drift term captures differences in deterministic temporal tendencies, while the diffusion term captures discrepancies in stochastic cross-variable coupling. We prove our greedy selection provides an additive $k$-center coverage guarantee after entropy-score normalization in Appendix A.

### 4.2. Geometric Metric Adaptation

As illustrated in Figure 1, given the input series $\boldsymbol{X} \in \mathbb{R}^{C \times T}$, we first utilize the TSFM backbone (e.g., Moirai 2.0) to

generate the model prediction $\widehat{\boldsymbol{X}}^{\text{model}} \in \mathbb{R}^{C \times T'}$. Subsequently, we transform $\widehat{\boldsymbol{X}}^{\text{model}}$ into compact channel embeddings to facilitate inter-channel dependency modeling. Given that the prediction horizon $T'$ is task-dependent (e.g., $T' \in \{96, 192, 336, 720\}$), a horizon-agnostic encoding strategy is essential. In contrast to naive global pooling, which often discards crucial temporal cues (Lee et al., 2021), we propose an adaptive trajectory projection to preserve the temporal patterns. Formally, for the $i$-th channel forecast $\widehat{\boldsymbol{X}}_i^{\text{model}} \in \mathbb{R}^{T'}$, we generate a horizon-agnostic representation $\boldsymbol{H}_i \in \mathbb{R}^D$ via a transformation:

$$\boldsymbol{H}_i = \text{MLP}(\text{Adaptive-AvgPool}(\widehat{\boldsymbol{X}}_i^{\text{model}}, S)), \quad (3)$$

where $\text{Adaptive-AvgPool}(\cdot, S)$ partitions the variable-length sequence into $S = 16$ temporal bins and averages within each bin, ensuring that the $k$-th element consistently aligns with the relative temporal position. The MLP then projects this aligned trajectory into the $D$-dimensional shared latent space, serving as the channel embedding that encodes the overall temporal patterns of the predicted curve.

Although $\mathbf{H}_i$ effectively encodes the temporal patterns of individual channels, reading inter-channel dependencies off these learned embeddings through a fixed dot-product similarity can be geometrically mismatched. The reason is that different datasets are governed by domain-specific diffusion terms $\mathbf{g}(\mathbf{x}_t; \boldsymbol{\Theta}_{\text{spat}})$, which control the stochastic intensity of variable interactions and therefore reshape how the

learned channel embeddings are distributed in space. Concretely, the diffusion term injects stochastic energy into the cross-variable coupling, so the same embedding distance no longer corresponds to the same dependency strength across domains. In high-diffusion domains (e.g., financial markets), this stochastic energy pushes the embeddings of even genuinely coupled channels far apart, producing an "inflated" embedding space where a large distance between two channel embeddings reflects noise rather than true inter-channel independence (Cont, 2001). In low-diffusion domains (e.g., power grids), the stable coupling keeps the embeddings tightly packed in a "compressed" space, where even a small distance between two channel embeddings can already signal a critical inter-channel causal link (Kundur, 2007). Hence an identical embedding distance encodes drastically different inter-channel dependency strengths depending on the physical context (Coifman et al., 2005). Consequently, applying a single universal similarity rule over these embeddings distorts the inferred dependencies. This issue is in fact amplified in prevailing Channel-Dependent (CD) approaches, which feed all channels into a dense attention module and learn cross-channel weights under one fixed similarity geometry; the attention is then misled into attending to channels that are merely close in the inflated/compressed embedding space rather than truly related, learning spurious correlations driven by domain-specific physical scales rather than genuine inter-channel causality.

To resolve this geometric conflict, we do not seek a forced projection into a static space, but rather construct a domain-adaptive Riemannian metric that aligns with the SDE-based underlying diffusion dynamics. Formally, such a metric defines a Mahalanobis-type distance $d_{\mathcal{D}}^2(\mathbf{H}_i, \mathbf{H}_j) = (\mathbf{H}_i - \mathbf{H}_j)^\top \mathbf{M}_{\mathcal{D}}(\mathbf{H}_i - \mathbf{H}_j)$ over the learned channel embeddings, where the Mahalanobis metric matrix $\mathbf{M}_{\mathcal{D}} \in \mathbb{R}^{D \times D}$ reweights the embedding space according to the diffusion geometry of domain $\mathcal{D}$. However, learning a full-rank metric tensor $\mathbf{M}_{\mathcal{D}}$ for each domain is computationally prohibitive in high-dimensional settings. We therefore restrict $\mathbf{M}_{\mathcal{D}}$ to a diagonal form and leverage the empirical physical statistics $\phi(\mathbf{X})$ as a compact domain signature to directly generate the adapted diagonal metric weights via a Hyper-Network. Before instantiating it, we first establish that a domain-specific metric is not merely a design choice but a theoretical necessity: no single fixed metric can stay accurate across domains with heterogeneous diffusion geometries. We formalize this impossibility in the following theorem.

**Theorem 4.3** (Universal Metric Error Bound). *Let $\mathcal{D}_1, \mathcal{D}_2$ be two domains with stationary diffusions and covariance proxies $\mathbf{\Sigma}_1, \mathbf{\Sigma}_2$, and let $\hat{\mathcal{G}}_{\mathbf{M}}^{(d)}$ and $\mathcal{G}^{*(d)}$ denote the inferred and population-optimal dependency graphs in $\mathcal{D}_d$ under a fixed-metric similarity rule, respectively. There exist constants $c > 0, \eta \geq 0$ such that for any fixed metric $\mathbf{M} \succeq \mathbf{0}$, $\|\hat{\mathcal{G}}_{\mathbf{M}}^{(1)} - \mathcal{G}^{*(1)}\|_{\mathrm{F}} + \|\hat{\mathcal{G}}_{\mathbf{M}}^{(2)} - \mathcal{G}^{*(2)}\|_{\mathrm{F}} \geq c \cdot \|\mathbf{\Sigma}_1 - \mathbf{\Sigma}_2\|_{\mathrm{F}} - \eta$.*

Theorem 4.3 establishes that metric distortion is mathematically inevitable under a fixed similarity metric across heterogeneous domains. To resolve this theoretical bottleneck, we propose the Geometric Hyper-Network ($\mathcal{H}_\theta$), which dynamically generates the domain-adaptive metric $\mathbf{M}_{\mathcal{D}}$ on-the-fly. Specifically, $\mathcal{H}_\theta$ takes the 7-dimensional feature vector $\phi(\mathbf{X})$ as domain context, and outputs the corresponding metric parameters:

$$[\boldsymbol{\gamma}_{\mathcal{D}}, \boldsymbol{\beta}_{\mathcal{D}}] = \mathcal{H}_\theta(\phi(\mathbf{X})), \quad \text{where } \mathcal{H}_\theta : \mathbb{R}^7 \to \mathbb{R}^{2D}. \quad (4)$$

In this metric relativity framework, the outputs have distinct physical interpretations: $\boldsymbol{\beta}_{\mathcal{D}}$ performs reference frame adjustment to neutralize domain-specific drift, while the diagonal vector $\boldsymbol{\gamma}_{\mathcal{D}}$ acts as adaptive metric weights. Based on Eq.(3), we construct the metric space embedding:

$$\boldsymbol{Z}_i = \mathrm{diag}(\boldsymbol{\gamma}_{\mathcal{D}})(\boldsymbol{H}_i - \boldsymbol{\beta}_{\mathcal{D}}) \in \mathbb{R}^D. \quad (5)$$

The goal here is to obtain a domain-adaptive metric over the channel embeddings; rather than materializing a metric tensor, we equivalently learn a rectified embedding $\boldsymbol{Z}_i$ in which a plain Euclidean distance already realizes this metric. Concretely, $\boldsymbol{Z}$ yields a dynamic weighted Mahalanobis distance over the original embeddings: $d_{\mathcal{D}}^2(\boldsymbol{H}_i, \boldsymbol{H}_j) = (\boldsymbol{H}_i - \boldsymbol{H}_j)^\top \mathrm{diag}(\boldsymbol{\gamma}_{\mathcal{D}}^2)(\boldsymbol{H}_i - \boldsymbol{H}_j) = \|\boldsymbol{Z}_i - \boldsymbol{Z}_j\|_2^2$. By predicting $\boldsymbol{\gamma}_{\mathcal{D}}$ on-the-fly, the model assigns a specialized ruler to each sample, down-weighting noise-dominated dimensions in high-diffusion domains while amplifying subtle but critical signal dimensions in stable domains.

### 4.3. Channel-Adapter Dependency Modeling

Leveraging the rectified embeddings $\boldsymbol{Z}$ from Eq.(5), we adopt a Channel adapter paradigm to capture inter-variable dependencies. Unlike Channel-Independent (CI) methods that ignore cross-channel interactions, our CP approach adaptively aggregates only geometrically relevant neighbors, effectively filtering out spurious noise while preserving critical multivariate signals.

To realize CP in a zero-shot setting without relying on a prespecified graph, we induce the dependency directly from the domain-adaptive geometry. Formally, we construct the sparse adjacency matrix $\boldsymbol{A}$ (contain self-loops) by computing the pairwise affinities in the rectified metric space as:

$$\boldsymbol{A} = \mathrm{Softmax}\left(\mathrm{TopK}\left(\frac{\boldsymbol{Z}\boldsymbol{Z}^\top}{\sqrt{D}}, K_{\mathcal{D}}\right)\right), \quad (6)$$

where $\sqrt{D}$ is the standard scaling factor and $\mathrm{Softmax}(\cdot)$ is a row-wise softmax normalization. $\mathrm{TopK}(\cdot, K_{\mathcal{D}})$ keeps the $K_{\mathcal{D}}$ strongest connections for each channel while masking the rest. Rather than treating $K_{\mathcal{D}}$ as a fixed hyperparameter, we extend $\mathcal{H}_\theta$ to jointly predict the adaptive neighborhood:

$$K_{\mathcal{D}} = \lfloor 1 + (C - 2) \cdot \sigma(\mathcal{H}_\theta^{(K)}(\phi(\mathbf{X}))), \quad (7)$$

where $\sigma(\cdot)$ is the sigmoid bounding $K_{\mathcal{D}} \in [1, C-1]$. To bypass the quadratic complexity $\mathcal{O}(C^2D)$ of dense materialization, we implement Eq.(6) via Approximate Nearest Neighbor search (Malkov & Yashunin, 2018). This optimization reduces graph construction to $\mathcal{O}(CD \log C)$, ensuring efficient scalability for high-dimensional data.

Using the induced graph, we perform message passing on backbone forecasts. Since the channel count $C$ can vary across domains in zero-shot settings, we adopt a parameter-efficient, $C$-agnostic broadcasting aggregation to propagate inter-channel dependencies across the prediction horizon:

$$\boldsymbol{Z}_{\text{agg}} = \boldsymbol{A}\,\widehat{\boldsymbol{X}}^{\text{model}} \in \mathbb{R}^{C \times T'}. \tag{8}$$

A key design principle is to preserve the temporal modeling capability of the backbone TSFM while only injecting inter-channel corrections. Pre-trained TSFMs such as Moirai 2.0 have already learned sophisticated temporal patterns, including seasonality, trends, and complex dynamics, from large-scale pre-training. Applying a naive linear projection would re-parameterize the temporal dimension, potentially distorting these well-calibrated predictions (Table 5). To address this, we adopt a gated residual mechanism that decomposes the refinement into two complementary components:

$$\boldsymbol{\delta} = f_\delta([\widehat{\boldsymbol{X}}^{\text{model}}; \boldsymbol{Z}_{\text{agg}}]), \ \ \mathbf{g} = \sigma\big(f_g([\widehat{\boldsymbol{X}}^{\text{model}}; \boldsymbol{Z}_{\text{agg}}])\big), \quad (9)$$

where $[\cdot\,;\cdot]$ denotes concatenation, $f_\delta$ and $f_g$ are lightweight MLPs projecting from $\mathbb{R}^{2T'}$ to $\mathbb{R}^{T'}$. Specifically, $f_\delta$ acts as a Correction Calculator, deriving the adjustment vector $\boldsymbol{\delta}$ by contrasting the backbone prediction with the global context $\boldsymbol{Z}_{\text{agg}}$. To prevent over-correction, $f_g$ serves as a Confidence Gate, outputting a score $\mathbf{g} \in (0, 1)$ that adaptively modulates the correction intensity.

The final prediction is obtained via gated addition:

$$\widehat{\boldsymbol{X}}^{\text{predict}} = \widehat{\boldsymbol{X}}^{\text{model}} + \mathbf{g} \odot \boldsymbol{\delta}, \tag{10}$$

where $\odot$ denotes element-wise (Hadamard) multiplication.

**Training Objective.** We train the plugin by minimizing the MSE prediction loss:

$$\mathcal{L}_{\text{MSE}} = \frac{1}{CT'} \sum_{i=1}^{C} \sum_{t=1}^{T'} \big\| \widehat{\boldsymbol{X}}^{\text{predict}}_{i,t} - \boldsymbol{X}^{\text{true}}_{i,t} \big\|_2^2, \quad (11)$$

where $\boldsymbol{X}^{\text{true}}$ denotes the ground-truth trajectory.

### 4.4. Computational Analysis

Our method achieves near-linear scalability with a time complexity of $\mathcal{O}(CD \log C + CKT')$ and a space complexity of $\mathcal{O}(CD + CK + CT')$, where $C$ is the number of channels, $K$ is the neighborhood size, $D$ is the embedding dimension, and $T'$ is the prediction horizon (see Appendix B).

## 5. Experiments

**Datasets.** We first aggregate a candidate pool of 1.2T observations from LOTSA, ECG, ERA5-Land, and Time-300B (Table 6). Applying our data selection strategy (Eq. (1)), we curate a representative 10% subset. For evaluation, we employ 9 diverse benchmarks covering meteorology, electricity, energy, transportation, and finance (Wu et al., 2023; Zhou et al., 2021; Lai et al., 2018). Please refer to Appendix A for more details about all adopted datasets.

**Backbones and Baselines.** To demonstrate model-agnostic versatility, we employ four representative TSFMs as backbones, including encoder-only Moirai 2.0 (Woo et al., 2024), decoder-only TimesFM (Das et al., 2024), encoder-decoder Chronos (Ansari et al., 2024), and MLP-based TTMs (Ekambaram et al., 2024). Given the scarcity of existing zero-shot adapters designed for MTSF, we benchmark against diverse SOTA full-shot baselines, including (1) Full fine-tuning methods: PCD (Lee et al., 2024) and TTM (Ekambaram et al., 2024), (2) Parameter-Efficient Fine-Tuning methods: PEFT (Beichter et al., 2025), Gen-PT (Liu et al., 2024b), AdaPTS (Benechehab et al., 2025); (3) In-context learning ICM (Żukowska et al., 2024). Appendix C list more details.

**Implementation Details.** For a fair evaluation, we reproduce all baselines with their official code and report their best performance. All experiments are conducted using PyTorch on a single NVIDIA RTX A100 80GB GPU. We use Adamw optimizer (Kingma & Ba, 2014) with learning rates selected from {1e-4, 1e-3}. The results are averaged over five runs with different random seeds. To prioritize pre-trained priors, we initialize the output weights of the residual branch $f_\delta$ to zero and the gate bias of $f_g$ to a negative value. This ensures that optimization starts from $\widehat{\boldsymbol{X}}^{\text{predict}} \approx \widehat{\boldsymbol{X}}^{\text{model}}$, facilitating gradual and stable adaptation without catastrophic interference. Please refer to Appendix D for more details.

### 5.1. Zero-shot Forecasting Performance

As shown in Tables 2 and 3, our extensive evaluation across nine diverse real-world datasets demonstrates that ChaTSFM consistently enhances the zero-shot forecasting capabilities of four TSFMs. The results yield three interesting findings. (1) **Performance gains correlate with physical coupling.** We observe that the improvement magnitude is intrinsically linked to the intensity of inter-variable physical correlations. In highly coupled domains like Solar and Traffic, ChaTSFM delivers substantial gains since solar stations exhibit synchronized fluctuations driven by shared solar irradiance, while traffic sensors capture causal flow dynamics. (2) **Spatial and temporal capabilities are orthogonal.** Our experiments indicate that the "spatial increment" provided by ChaTSFM is decoupled from

*Table 2.* Zero-shot performance after applying our proposed ChaTSFM to TSFMs. **The input length is fixed at 96, and the forecasting horizons are 96, 192, 336, 720.** We report the mean MSE/MAE averaged over the four horizons. Best results are highlighted in **bold**. * denotes TSFMs whose pre-training data may overlap with the evaluation datasets, leading to potential data leakage.

| Method | Moirai | | +ChaTSFM | | TimesFm | | +ChaTSFM | | Chronos | | +ChaTSFM | | TTMs | | +ChaTSFM | | —Avg. Imp.— | |
|---|---|---|---|---|---|---|---|---|---|---|---|---|---|---|---|---|---|---|
| | MSE | MAE | MSE | MAE | MSE | MAE | MSE | MAE | MSE | MAE | MSE | MAE | MSE | MAE | MSE | MAE | MSE | MAE |
| ETTh1 | 0.508 | 0.444 | 0.496 | 0.440 | 0.583 | 0.477 | 0.569 | 0.472 | 0.524 | 0.443 | 0.512 | 0.439 | 0.396 | 0.419 | **0.386** | **0.414** | 2.39% | 1.01% |
| ETTh2 | 0.476 | 0.436 | 0.444 | 0.428 | 0.458 | 0.427 | 0.427 | 0.420 | 0.424 | 0.411 | 0.396 | 0.403 | 0.364 | 0.396 | **0.341** | **0.388** | 6.60% | 1.86% |
| ETTm1 | 0.950 | 0.609 | 0.808 | 0.536 | 0.667 | 0.512 | 0.567 | 0.451 | 1.032 | 0.587 | 0.677 | 0.519 | 0.390 | 0.389 | **0.334** | **0.343** | 19.67% | 11.82% |
| ETTm2 | 0.383 | 0.386 | 0.343 | 0.376 | 0.376 | 0.377 | 0.343 | 0.368 | 0.351 | 0.366 | 0.320 | 0.357 | 0.287 | 0.329 | **0.260** | **0.321** | 9.36% | 2.46% |
| Exchange | 0.338 | 0.390 | **0.318** | **0.380** | 0.381* | 0.420* | 0.359 | 0.410 | 0.369 | 0.407 | 0.348 | 0.397 | 0.386 | 0.408 | 0.363 | 0.398 | 5.83% | 2.46% |
| Solar | 0.515 | 0.399 | 0.394 | 0.318 | 0.735 | 0.510 | 0.574 | 0.411 | 1.922 | 0.908 | 1.496 | 0.733 | 0.214 | 0.265 | **0.166** | **0.212** | 22.49% | 19.74% |
| Weather | 0.301 | 0.302 | 0.255 | 0.295 | 0.200* | 0.216* | **0.169** | **0.212** | 0.412 | 0.331 | 0.355 | 0.324 | 0.229 | 0.263 | 0.195 | 0.257 | 14.86% | 2.14% |
| Electricity | 0.224 | 0.294 | 0.211 | 0.286 | 0.222 | 0.300 | 0.210 | 0.294 | 0.217* | 0.283* | 0.206 | **0.277** | 0.211 | 0.300 | **0.200** | 0.293 | 5.37% | 2.29% |
| Traffic | 0.566* | 0.320* | 0.520 | 0.303 | 0.503* | 0.284* | **0.463** | **0.270** | 0.738 | 0.337 | 0.680 | 0.321 | 0.560 | 0.364 | 0.516 | 0.346 | 7.94% | 4.98% |

the backbone's temporal capability. Despite vast baseline differences on Solar, ChaTSFM consistently reduces errors across all architectures. This suggests that ChaTSFM acts as an independent module yielding additive information gain regardless of the backbone's inherent temporal strength. (3) **Robustness to extended context windows.** As shown in Table 3, ChaTSFM maintains superiority even when the input length extends to 3,072. This demonstrates that simply extending historical context cannot fully substitute for explicit spatial modeling, validating the noise-robustness of our sparse aggregation mechanism.

*Table 3.* Zero-shot Performance via longer input size. We use four prediction horizons {96, 192, 336, 720} with the corresponding input time series lengths {512, 1024, 2048, 3072}.

| Dataset | Moirai 2.0 | | +ChaTSFM | | Chronos | | +ChaTSFM | |
|---|---|---|---|---|---|---|---|---|
| | MSE | MAE | MSE | MAE | MSE | MAE | MSE | MAE |
| ETTh1 | 0.428 | 0.427 | 0.418 | 0.422 | 0.591 | 0.468 | 0.577 | 0.461 |
| ETTh2 | 0.361 | 0.384 | 0.339 | 0.378 | 0.405 | 0.410 | 0.380 | 0.402 |
| ETTm1 | 0.436 | 0.430 | 0.368 | 0.386 | 0.645 | 0.500 | 0.561 | 0.455 |
| ETTm2 | 0.307 | 0.347 | 0.280 | 0.338 | 0.310 | 0.350 | 0.285 | 0.340 |
| Weather | 0.275 | 0.286 | 0.241 | 0.280 | 0.292 | 0.315 | 0.257 | 0.308 |

## 5.2. Full-Shot Forecasting Performance

As shown in Table 4, ChaTSFM consistently outperforms all adapter baselines under the full-shot setting, with the margin being especially pronounced on multivariate-intensive datasets such as Solar and Traffic, where dense inter-channel coupling dominates the forecasting signal. On Solar, for instance, ChaTSFM reduces the error of the Moirai backbone by over 50%, substantially surpassing the strongest competing adapter, while on weakly-coupled datasets such as Exchange the improvement is comparatively modest. This contrast confirms that our gains stem from explicitly modeling spatial structure rather than from temporal overfitting, and is driven by two complementary mechanisms: **(1) Transferable geometric priors.** Across datasets, the full ChaTSFM consistently improves over its w/o Pre-train variant, indicating that the metric calibration and dependency patterns learned during pre-training are not dataset-specific

but transfer across heterogeneous domains. Because the curated corpus exposes the plugin to a wide spectrum of drift–diffusion regimes, the Hyper-Network learns a generic mapping from domain statistics to metric geometry, rather than memorizing any single topology. In full-shot fine-tuning, these priors act as a better geometric initialization: instead of searching for a sensible inter-channel metric from scratch, optimization starts from a near-correct geometry, which leads to faster and more stable convergence and consistently lower final error. **(2) Channel-partial inductive bias.** Notably, even w/o Pre-train already outperforms most baselines, showing that a large portion of the gain is architectural rather than data-driven. We attribute this to two design choices. First, Top-$K$ sparse aggregation restricts each channel to attend only to its most relevant neighbors, mitigating the noise that dense channel mixing introduces when many channels are spuriously correlated—an effect that is amplified on high-dimensional datasets like Traffic (862 channels). Second, the gated residual fusion injects inter-channel context only when it yields information gain, leaving the backbone's well-calibrated temporal predictions intact elsewhere. Together, these two biases let the model exploit cross-channel synergies without sacrificing the temporal structure inherited from the frozen TSFM, explaining why the gains are robust across both strongly- and weakly-coupled regimes.

## 5.3. Efficiency Analysis

Figure 2 compares different methods from three aspects: forecasting accuracy, training/adaptation time, and storage footprint. The left panel shows that ChaTSFM lies in the low-error and low-cost region, achieving a favorable efficiency–performance trade-off on ETTm1. Compared with large TSFMs such as Moirai, Chronos, and TimesFM, ChaTSFM avoids costly full-model adaptation and introduces only a lightweight plugin. Its 48K trainable parameters correspond to about 190 KB under FP32 storage, which is orders of magnitude smaller than the storage footprint of large TSFMs. Compared with lightweight baselines, ChaTSFM achieves lower MSE while maintaining compa-

*Table 4.* Full-shot performance comparison among adapters. The input length is 96, and the forecasting horizons are {96, 192, 336, 720}. The best and second-best average performance are highlighted in **bold** and underlined. w/o Pre-train denotes +ChaTSFM without pre-training. * denotes TSFMs whose pre-training data may overlap with the evaluation datasets, leading to potential data leakage.

| Method | Moirai 2.0$_{small}$ | | +ICM | | +PCD | | +PEFT | | +Gen-PT | | +AdaPTS | | +TTM | | +ChaTSFM | | w/o Pre-train | |
|---|---|---|---|---|---|---|---|---|---|---|---|---|---|---|---|---|---|---|
| | MSE | MAE | MSE | MAE | MSE | MAE | MSE | MAE | MSE | MAE | MSE | MAE | MSE | MAE | MSE | MAE | MSE | MAE |
| ETTh1 | 0.508 | 0.444 | 0.504 | 0.447 | 0.502 | 0.445 | 0.500 | 0.446 | 0.500 | 0.444 | 0.497 | 0.444 | 0.497 | 0.445 | **0.485** | **0.442** | 0.492 | 0.444 |
| ETTh2 | 0.476 | 0.436 | 0.469 | 0.440 | 0.467 | 0.439 | 0.465 | 0.438 | 0.462 | 0.437 | 0.460 | 0.436 | 0.458 | 0.435 | **0.448** | **0.432** | 0.452 | 0.434 |
| ETTm1 | 0.950 | 0.609 | 0.824 | 0.583 | 0.810 | 0.578 | 0.795 | 0.571 | 0.772 | 0.562 | 0.733 | 0.553 | 0.687 | 0.540 | **0.621** | **0.518** | 0.642 | 0.524 |
| ETTm2 | 0.383 | 0.386 | 0.363 | 0.388 | 0.363 | 0.388 | 0.359 | 0.388 | 0.354 | 0.384 | 0.356 | 0.384 | 0.352 | 0.383 | **0.339** | **0.377** | 0.344 | 0.380 |
| Exchange | 0.338 | 0.390 | 0.330 | 0.388 | 0.329 | 0.388 | 0.327 | 0.388 | 0.327 | 0.385 | 0.325 | 0.385 | 0.324 | 0.385 | **0.321** | **0.382** | 0.323 | 0.384 |
| Solar | 0.515 | 0.399 | 0.334 | 0.332 | 0.333 | 0.328 | 0.306 | 0.328 | 0.287 | 0.321 | 0.271 | 0.312 | 0.262 | 0.313 | **0.228** | **0.287** | 0.242 | 0.299 |
| Weather | 0.301 | 0.302 | 0.285 | 0.302 | 0.284 | 0.301 | 0.280 | 0.301 | 0.274 | 0.299 | 0.269 | 0.299 | 0.270 | 0.298 | **0.258** | **0.298** | 0.262 | 0.299 |
| Electricity | 0.224 | 0.294 | 0.214 | 0.293 | 0.212 | 0.294 | 0.211 | 0.291 | 0.207 | 0.289 | 0.206 | 0.287 | 0.206 | 0.288 | **0.199** | **0.284** | 0.203 | 0.286 |
| Traffic | 0.566* | 0.320* | 0.542 | 0.313 | 0.539 | 0.310 | 0.538 | 0.311 | 0.533 | 0.309 | 0.530 | 0.308 | 0.528 | 0.306 | **0.517** | **0.302** | 0.523 | 0.304 |

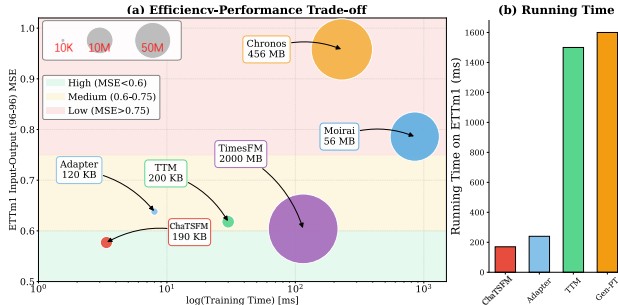

*Figure 2.* Efficiency–performance–footprint trade-off on ETTm1. The x-axis reports the per-iteration training/adaptation time, and the y-axis reports the forecasting MSE, where lower values indicate better accuracy. Bubble size and text labels indicate the FP32 storage footprint of the corresponding model.

rable or lower runtime, as shown in the right panel. These results demonstrate that the proposed sparse adaptation design improves forecasting accuracy with only minimal computational and storage overhead.

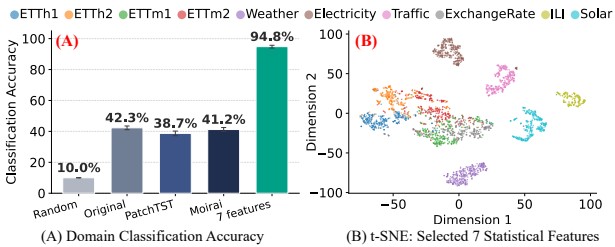

*Figure 3.* Validation of Generative Proxies. (A) Linear classification comparison of baselines and (B) distinct t-SNE visualization.

## 5.4. Domain Representation Analysis

To validate the discriminative power of the generative proxies in Section 4.1, we evaluate via a 10-way classification in Figure 3. We benchmark our compact statistics against Raw Data and Deep Predictive Outputs (from Moirai 2.0 and PatchTST), using a shared logistic regression classifier to isolate representational quality. Our representation

achieves 94.8% accuracy (Panel A), outperforming all baselines. This is confirmed by the compact, well-separated clusters in Panel (B). These results confirm that the proposed drift-diffusion statistics successfully capture the intrinsic physical signature of each domain. More details in Appendix E.

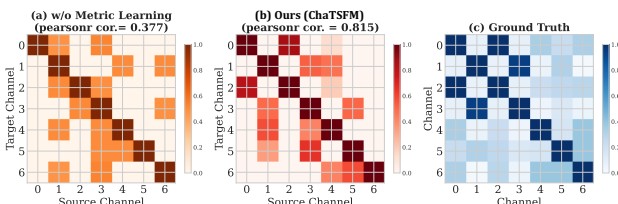

*Figure 4.* Learned channel dependencies at the 48-th forecasting step on ETTm1: (a) w/o metric learning, (b) Ours, (c) ground-truth.

## 5.5. Metric Adaptation Analysis

We validate geometric alignment (Section 4.2) by visualizing the learned dependencies in Figure 4. In Panel (a), w/o metric learning, the model relies on raw embeddings and fails to capture meaningful dependencies, resulting in a noisy attention map. Quantitatively, ChaTSFM yields high alignment with physical reality (r=0.815), effectively suppressing noise. This validates Theorem 4.3 as ChaTSFM overcomes the geometric distortion of the raw embeddings, allowing the model to capture statistically valid channel correlations.

## 5.6. Ablation Study

As shown in Table 5, **(1) w/o Dataset Selection:** Random sampling, single-objective, and full dataset degrade performance, confirming that naively increasing data volume introduces redundancy and noise, whereas ChaTSFM balances informativeness and diversity to curate a high-quality pre-training corpus. **(2) w/o Metric Learning:** We benchmark against Learnable Global, which applies universal metric parameters, and Instance-wise MLP, which generates param-

*Table 5.* Ablation study of ChaTSFM. Look-back horizon is 96 and results are averaged over forecasting horizons $\{96, 192, 336, 720\}$.

| Variants | | ETTh1 | | ETTh2 | | Exchange | | Solar | | Weather | | Electricity | | Traffic | |
|---|---|---|---|---|---|---|---|---|---|---|---|---|---|---|---|
| | | MSE | MAE | MSE | MAE | MSE | MAE | MSE | MAE | MSE | MAE | MSE | MAE | MSE | MAE |
| w/o Data Selection | Random Selection | 0.501 | 0.443 | 0.463 | 0.434 | 0.331 | 0.387 | 0.435 | 0.338 | 0.284 | 0.299 | 0.220 | 0.291 | 0.549 | 0.310 |
| | Max Entropy Only | 0.499 | 0.442 | 0.455 | 0.433 | 0.329 | 0.385 | 0.408 | 0.329 | 0.272 | 0.298 | 0.218 | 0.290 | 0.537 | 0.305 |
| | Max Diversity Only | 0.498 | 0.441 | 0.452 | 0.431 | 0.325 | 0.384 | 0.405 | 0.328 | 0.267 | 0.297 | 0.216 | 0.289 | 0.533 | 0.305 |
| | Full Dataset | 0.502 | 0.443 | 0.466 | 0.434 | 0.333 | 0.387 | 0.444 | 0.344 | 0.298 | 0.301 | 0.222 | 0.292 | 0.551 | 0.312 |
| w/o Metric | Cosine Similarity | 0.505 | 0.443 | 0.475 | 0.435 | 0.335 | 0.387 | 0.512 | 0.396 | 0.298 | 0.300 | 0.222 | 0.293 | 0.562 | 0.318 |
| | Learnable Global | 0.506 | 0.444 | 0.474 | 0.434 | 0.334 | 0.388 | 0.511 | 0.395 | 0.296 | 0.299 | 0.223 | 0.292 | 0.564 | 0.318 |
| | Instance-wise MLP | 0.502 | 0.441 | 0.455 | 0.432 | 0.328 | 0.385 | 0.418 | 0.331 | 0.268 | 0.297 | 0.219 | 0.289 | 0.536 | 0.305 |
| w/o Adapter | Channel Independent | 0.508 | 0.444 | 0.476 | 0.436 | 0.338 | 0.390 | 0.515 | 0.399 | 0.301 | 0.302 | 0.224 | 0.294 | 0.566 | 0.320 |
| | Full Attention | 0.502 | 0.443 | 0.466 | 0.434 | 0.333 | 0.387 | 0.435 | 0.338 | 0.284 | 0.299 | 0.220 | 0.291 | 0.550 | 0.311 |
| | Fixed Top-5 | 0.499 | 0.442 | 0.448 | 0.431 | 0.322 | 0.385 | 0.402 | 0.327 | 0.271 | 0.299 | 0.219 | 0.291 | 0.534 | 0.306 |
| w/o Gate | Direct Addition | 0.502 | 0.443 | 0.464 | 0.434 | 0.331 | 0.387 | 0.440 | 0.340 | 0.286 | 0.300 | 0.220 | 0.291 | 0.552 | 0.312 |
| | Linear Residual | 0.503 | 0.443 | 0.468 | 0.434 | 0.335 | 0.388 | 0.448 | 0.346 | 0.290 | 0.300 | 0.221 | 0.292 | 0.554 | 0.313 |
| Moirai 2.0+ChaTSFM | | **0.496** | **0.440** | **0.444** | **0.428** | **0.318** | **0.380** | **0.394** | **0.318** | **0.255** | **0.295** | **0.211** | **0.286** | **0.520** | **0.303** |

eters from individual samples. The performance drops with these alternatives confirm the necessity of domain-adaptive metrics for correcting geometric distortion. **(3) w/o Channel Adapter:** Performance drops in CI, Full Attn, and Fixed Top-5 expose the limits of missing dependencies, noise, and restricted receptive fields, respectively. ChaTSFM effectively identifies the optimal sparse geometric structure, ensuring robust spatial modeling without noise. **(4) w/o Gated Refinement:** Removing the gating mechanism (Direct Addition) or using Linear Residuals degrades performance, confirming that our gate selectively injects spatial context only where it offers valid information gain.

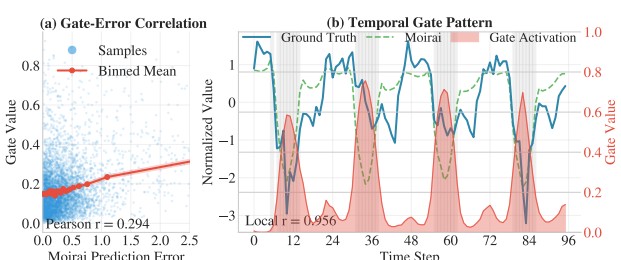

*Figure 5.* (a) Positive correlation between gate values and prediction errors. (b) Gate activates precisely at high-error time steps.

### 5.7. Gated Refinement Analysis

To verify the adaptive correction mechanism (Section 4.3), Figure 5 analyzes the learned gating behavior on ETTh1. Panel (a) reveals a distinct positive correlation (r=0.30) between the **gate magnitude** and **absolute prediction error**: as the backbone error grows, the binned-mean gate value rises monotonically, indicating that ChaTSFM automatically intensifies corrections for hard-to-predict samples while staying near-dormant on easy ones. At the instance level in Panel (b), the gate stays close to zero where the backbone already tracks the ground truth and spikes precisely at the high-error time steps (local $r$=0.96). Consequently, the mechanism acts as a surgical filter, preserving reliable tem-

poral priors while exclusively targeting high-error regions for refinement.

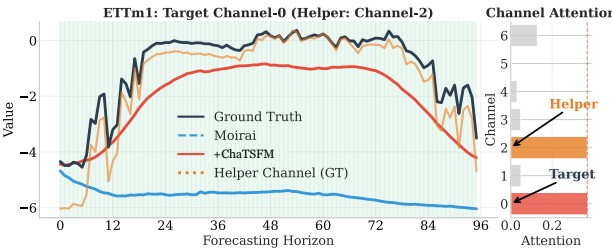

*Figure 6.* **Case Study.** Leveraging inter-channel dependencies enables ChaTSFM to correct the baseline and recover the plateau.

### 5.8. Case Study: Adaptive Inter-Channel Aggregation

Figure 6 visualizes a case where the CI-based Moirai 2.0 fails to capture a critical plateau trend (Blue). In the right panel, ChaTSFM identifies the underlying dependency, assigning dominant attention weights to the pivotal "helper" (Channel-2) while suppressing noise. By using this highly correlated auxiliary signal, ChaTSFM (Red) successfully rectifies the deviation and reconstructs the plateau trajectory that Moirai 2.0 misses. This exemplifies how our model effectively aggregates multivariate context to enhance zero-shot performance.

## 6. Conclusion

In this work, we propose ChaTSFM, a lightweight adapter that injects multivariate dependencies into TSFMs. By leveraging domain-aware channel adaptation, ChaTSFM significantly boosts forecasting accuracy with minimal training overhead. Our findings reveal a fundamental insight: effective reasoning in TSFMs hinges on geometric coherence, not merely parameter scale. We believe ChaTSFM paves the way for grounded universal time series intelligence.

## Impact Statement

This paper presents work whose goal is to advance the field of Machine Learning, specifically in zero-shot multivariate time series forecasting. Our proposed framework, ChaTSFM, improves the ability of foundation models to model complex real-world systems, such as power grids, traffic networks, and weather patterns. By improving the accuracy of the forecast in these domains, our work has the potential to contribute to more efficient integration of renewable energy, optimized transportation logistics, and better resource allocation. Furthermore, as a lightweight plugin that requires minimal training overhead, ChaTSFM democratizes access to high-performance forecasting and reduces the carbon footprint associated with training models from scratch. Although our method improves zero-shot generalization, the deployment of AI forecasting in critical infrastructure carries inherent risks. We emphasize that such models should serve as decision-support tools with appropriate human oversight and safety redundancies, rather than autonomous decision-makers in high-stakes scenarios.

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

# A. Dataset

## A.1. Pre-training Dataset Selection

Building upon the domain fingerprinting established in Sec. 4.1, we implement a rigorous data curation strategy to ensure robust zero-shot generalization. Our objective is to identify a representative subset from a massive candidate pool that maximizes generative diversity while prioritizing complex multivariate topologies, all within a limited computational budget.

**Dataset Pool Construction.** As shown in Table 6, we assemble a comprehensive candidate pool by aggregating large-scale time series collections, including LOTSA (Woo et al., 2024), ECG (Goldberger et al., 2000), ERA5-Land (Muñoz-Sabater et al., 2021), and Time-300B (Shi et al., 2024). These sources encompass a vast spectrum of heterogeneous physical domains, providing the necessary variance in drift and diffusion patterns required for training our plugin.

**Exclusion Criteria.** We deliberately exclude datasets restricted to univariate sequences, such as Chronos (Ansari et al., 2024) and Timer (Liu et al., 2024d). These collections are tailored for the Channel-Independence (CI) paradigm, which inherently discards the inter-channel synergies ChaTSFM aims to unlock. Since our primary objective is to resolve geometric distortions in multivariate modeling, CI-based datasets offer limited utility for inducing domain-aware topologies.

*Table 6.* Comparison of candidate dataset sources. The curated pool is constructed to maximize coverage across generative meta-domains, ensuring a diverse spectrum of drift and diffusion patterns while strictly preserving inter-channel topologies for multivariate modeling.

| Attribute | LOTSA | ECG | ERA5-Land | Time-300B | Combined Pool |
|---|---|---|---|---|---|
| Total Size (Tokens) | 230B | 48B | 643B | 309B | **1.2T** |
| # Subsets | 174 | 6 | 6 | 107 | 230 |
| # Domains | 9 | 1 | 1 | 9 | 10 |
| Multivariate | Partial | ✓ | ✓ | Partial | Partial |

Guided by the selection strategy established in Sec. 4.1, we curate a pre-training corpus as summarized in Table 7. These datasets are specifically selected for their heterogeneous drift-diffusion profiles, which ensure that the plugin encounters a comprehensive spectrum of multivariate interaction topologies during training. The selection process is governed by a greedy curation algorithm (detailed in Algorithm 1), which balances the statistical informativeness of each domain with its generative diversity. Specifically, by maximizing the Gaussian $W_2$ distance between candidate Gaussian proxies and the existing subset $\mathcal{M}$, we ensure that ChaTSFM is exposed to diverse cross-variable coupling strengths. We prove that our greedy selection provides an additive $k$-center coverage guarantee after entropy-score normalization; the full theoretical derivation and pseudocode are provided in Section F.3.

---

**Algorithm 1** Greedy Generative Domain Selection (Eq. (1))

---

**Require:** Candidate pool $\mathcal{U} = \{\mathcal{D}_1, \ldots, \mathcal{D}_N\}$, budget $K$, weight $\lambda$, stabilizer $\epsilon$, metric $d(\mathcal{D}, \mathcal{D}') = W_2(\mathcal{D}, \mathcal{D}')$
**Ensure:** Selected set $\mathcal{M}$ with $|\mathcal{M}| = K$
  1: Estimate $(\boldsymbol{\mu}_{\mathcal{D}}, \boldsymbol{\Sigma}_{\mathcal{D}})$ for all $\mathcal{D} \in \mathcal{U}$          # Extract Gaussian proxies (drift and diffusion) for all domains
  2: $h(\mathcal{D}) \leftarrow \log \det(\boldsymbol{\Sigma}_{\mathcal{D}} + \epsilon\mathbf{I})$          # Compute raw entropy-based informativeness score
  3: $\bar{h}(\mathcal{D}) \leftarrow \text{Normalize}(h(\mathcal{D}))$          # Perform min-max normalization for scale alignment
  4: $\mathcal{M} \leftarrow \emptyset$          # Initialize the selected subset
  5: $\mathcal{D}^{(1)} \leftarrow \arg\max_{\mathcal{D} \in \mathcal{U}} \lambda \bar{h}(\mathcal{D})$          # Select the most informative domain as the starting seed
  6: $\mathcal{M} \leftarrow \mathcal{M} \cup \{\mathcal{D}^{(1)}\}$          # Update the selected subset
  7: **for** $t = 2$ to $K$ **do**
  8:      **for all** $\mathcal{D} \in \mathcal{U} \setminus \mathcal{M}$ **do**
  9:          $\Delta(\mathcal{D}; \mathcal{M}) \leftarrow \min_{\mathcal{D}_j \in \mathcal{M}} d(\mathcal{D}, \mathcal{D}_j)$          # Calculate the nearest-neighbor distance (diversity gain)
10:          $\text{SCORE}(\mathcal{D}) \leftarrow \lambda \bar{h}(\mathcal{D}) + \Delta(\mathcal{D}; \mathcal{M})$          # Balance informativeness and diversity via Eq. (1)
11:      **end for**
12:      $\mathcal{D}^{(t)} \leftarrow \arg\max_{\mathcal{D} \in \mathcal{U} \setminus \mathcal{M}} \text{SCORE}(\mathcal{D})$          # Select the candidate maximizing the joint utility
13:      $\mathcal{M} \leftarrow \mathcal{M} \cup \{\mathcal{D}^{(t)}\}$          # Iteratively update the selected subset
14: **end for**
15: **return** $\mathcal{M}$          # Output the final curated meta-domain subset

---

*Table 7.* Selected datasets and their statistics.

| Dataset | Domain | #Vars | #Series | Total Len | Avg Len | Frequency | Location |
|---|---|---|---|---|---|---|---|
| PEMS04 | Traffic | 3 | 307 | 5,216,544 | 16,992 | 5min | California, USA |
| residential_pv_power | Energy | 3 | 233 | 125,338,950 | 537,935 | hourly | Australia |
| cdc_fluview_ilinet | Health | 5 | 75 | 63,903 | 852 | weekly | USA |
| cmip6_1975 | Climate | 53 | 8192 | 59,801,600 | 7,300 | monthly | Global |
| cdc_fluview_who_nrevss | Health | 4 | 74 | 41,760 | 564 | weekly | USA |
| residential_load_power | Energy | 3 | 271 | 145,994,559 | 538,725 | hourly | Australia |
| china_air_quality | Environment | 6 | 437 | 5,739,234 | 13,133 | hourly | China |
| era5_1992 | Weather | 45 | 8192 | 71,565,312 | 8,736 | hourly | Global |
| alibaba_cluster_trace_2018 | Cloud | 2 | 58,409 | 95,192,530 | 1,630 | 10sec | Alibaba Cloud |
| beijing_air_quality | Environment | 11 | 12 | 420,768 | 35,064 | hourly | Beijing, China |
| subseasonal | Climate | 4 | 862 | 14,197,140 | 16,470 | weekly | Global |
| borg_cluster_data_2011 | Cloud | 2 | 143,386 | 537,552,854 | 3,749 | 5min | Google Cloud |
| PEMS08 | Traffic | 3 | 170 | 3,035,520 | 17,856 | 5min | California, USA |
| era5_2005 | Weather | 45 | 8192 | 71,565,312 | 8,736 | hourly | Global |
| cmip6_1985 | Climate | 53 | 8192 | 59,801,600 | 7,300 | monthly | Global |
| era5_2002 | Weather | 45 | 8192 | 71,565,312 | 8,736 | hourly | Global |
| cmip6_1935 | Climate | 53 | 8192 | 59,801,600 | 7,300 | monthly | Global |
| era5_2011 | Weather | 45 | 8192 | 71,565,312 | 8,736 | hourly | Global |
| era5_2013 | Weather | 45 | 8192 | 71,565,312 | 8,736 | hourly | Global |
| cmip6_1910 | Climate | 53 | 8192 | 59,801,600 | 7,300 | monthly | Global |
| cmip6_1890 | Climate | 53 | 8192 | 59,801,600 | 7,300 | monthly | Global |
| era5_2017 | Weather | 45 | 8192 | 71,565,312 | 8,736 | hourly | Global |
| cmip6_1980 | Climate | 53 | 8192 | 59,801,600 | 7,300 | monthly | Global |
| era5_1996 | Weather | 45 | 8192 | 71,565,312 | 8,736 | hourly | Global |
| cmip6_1965 | Climate | 53 | 8192 | 59,801,600 | 7,300 | monthly | Global |

## A.2. Evaluation Datasets

In Table 8, we summarize the detailed statistics of datasets used in our experiments. We conduct extensive evaluations on 9 widely used multivariate time series datasets for forecasting. The ETT dataset (Zhou et al., 2021) contains temperature and power load data collected at hourly (ETTh) and 15-minute (ETTm) intervals. Exchange (Lai et al., 2018) compiles daily exchange rate information from 1990 to 2016. Solar-Energy (Lai et al., 2018) provides 10-minute solar power readings from 137 PV plants. The Weather (Wu et al., 2023), Electricity (Wu et al., 2023), and Traffic (Wu et al., 2023) datasets include high-resolution environmental and usage signals, recorded respectively at 10-minute, hourly, and hourly intervals.

## B. Time and Space Complexity.

We provide a step-by-step complexity breakdown for the proposed "select–adapt–refine" pipeline. We report the per-instance cost (i.e., one multivariate series sample) and treat small constants (e.g., the number of domain statistics, pooling bins, and MLP layer widths) as $\mathcal{O}(1)$, since they are fixed across all experiments. Let $C$ be the number of channels, $T$ the input length, $T'$ the forecasting horizon, and $D$ the channel-embedding dimension. Let $K$ denote the (possibly domain-adaptive) neighborhood size after sparsification. We use $S$ to denote the number of pooling bins in Eq.(3) (fixed to $S = 16$ in our implementation), and let $\phi(\mathbf{X}) \in \mathbb{R}^7$ be the fixed-dimensional domain statistic vector. We analyze only the plugin overhead beyond the frozen TSFM backbone.

**Stage 0: Frozen TSFM forecasting.** The backbone produces $\widehat{\mathbf{X}}^{\text{model}} \in \mathbb{R}^{C \times T'}$. Since the backbone is fixed and model-dependent, we exclude its cost from the plugin complexity.

**Stage 1: Trajectory encoder (Eq.(3)).** For each channel $i$, AdaptiveAvgPool partitions a length-$T'$ sequence into $S$ bins and averages within each bin. This requires a single pass over $\widehat{\mathbf{X}}_i^{\text{model}}$, hence $\mathcal{O}(T')$ per channel. The subsequent MLP maps $\mathbb{R}^S \to \mathbb{R}^D$; with fixed depth and width, its per-channel cost is $\mathcal{O}(D)$. Overall,

$$\text{Time: } \mathcal{O}(CT' + CD), \qquad \text{Space: } \mathcal{O}(CD),$$

where the space term stores the channel embeddings $\mathbf{H} \in \mathbb{R}^{C \times D}$.

*Table 8.* Statistics of all datasets used in our experiments. "Frequency" denotes the sampling interval of time points.

| Dataset | Type | Dim | Lookback Length | Prediction Length | Total Length | Frequency | Domain |
|---|---|---|---|---|---|---|---|
| *Long-term Forecasting Datasets* | | | | | | | |
| ETTh1 | Multivariate | 7 | 96 | {96, 192, 336, 720} | 17,420 | Hourly | Electricity |
| ETTh2 | Multivariate | 7 | 96 | {96, 192, 336, 720} | 17,420 | Hourly | Electricity |
| ETTm1 | Multivariate | 7 | 96 | {96, 192, 336, 720} | 69,680 | 15-min | Electricity |
| ETTm2 | Multivariate | 7 | 96 | {96, 192, 336, 720} | 69,680 | 15-min | Electricity |
| Exchange | Multivariate | 8 | 96 | {96, 192, 336, 720} | 7,588 | Daily | Finance |
| Electricity | Multivariate | 321 | 96 | {96, 192, 336, 720} | 26,304 | Hourly | Energy |
| Weather | Multivariate | 21 | 96 | {96, 192, 336, 720} | 52,695 | 10-min | Meteorology |
| Solar | Multivariate | 137 | 96 | {96, 192, 336, 720} | 52,560 | 10-min | Energy |
| Traffic | Multivariate | 862 | 96 | {96, 192, 336, 720} | 17,544 | Hourly | Transportation |

**Stage 2: Geometric hyper-network and metric rectification (Eq.(5)).** The hyper-network $\mathcal{H}_\theta : \mathbb{R}^7 \to \mathbb{R}^{2D}$ produces $(\gamma_\mathcal{D}, \beta_\mathcal{D})$. Because the input dimension (7) and MLP depth are fixed, this cost is $\mathcal{O}(D)$ per instance. Applying the diagonal metric and shift to all channels costs $\mathcal{O}(CD)$: $\mathbf{Z}_i = \mathrm{diag}(\gamma_\mathcal{D}) (\mathbf{H}_i - \beta_\mathcal{D})$. Thus,

$$\text{Time: } \mathcal{O}(CD), \qquad \text{Space: } \mathcal{O}(CD),$$

where $\mathbf{Z} \in \mathbb{R}^{C \times D}$ is stored.

**Stage 3: Sparse topology induction with ANN (Eq.(6)).** A naive dense affinity computation $\mathbf{ZZ}^\top$ would cost $\mathcal{O}(C^2D)$, which is prohibitive for large $C$. Instead, we construct a *sparse $K$-NN graph* using Approximate Nearest Neighbor (ANN) search. Under standard ANN indexing/query schemes (e.g., HNSW), querying $K$ neighbors for each of the $C$ vectors in $\mathbb{R}^D$ requires approximately $\mathcal{O}(CD \log C)$ time.[1] We then apply a row-wise softmax only on the retained $K$ entries per row, costing $\mathcal{O}(CK)$. Therefore,

$$\text{Time: } \mathcal{O}(CD \log C + CK), \qquad \text{Space: } \mathcal{O}(CK),$$

where $\mathbf{A}$ is stored in sparse form (indices + values) with $CK$ nonzeros (including self-loops).

**Stage 4: Broadcasting aggregation on forecasts (Eq.(8)).** We compute $\mathbf{Z}_{\mathrm{agg}} = \mathbf{A}\widehat{\mathbf{X}}^{\mathrm{model}} \in \mathbb{R}^{C \times T'}$. Because $\mathbf{A}$ has $CK$ nonzeros, sparse-dense multiplication costs $\mathcal{O}(CKT')$ time and stores the aggregated forecasts $\mathbf{Z}_{\mathrm{agg}} \in \mathbb{R}^{C \times T'}$:

$$\text{Time: } \mathcal{O}(CKT'), \qquad \text{Space: } \mathcal{O}(CT').$$

**Stage 5: Gated residual refinement (Eqs.(9–10)).** For each channel, two lightweight MLPs $f_\delta$ and $f_g$ take a vector in $\mathbb{R}^{2T'}$ and output $\mathbb{R}^{T'}$. With fixed architecture, the cost scales linearly in the output length, leading to $\mathcal{O}(CT')$ time and $\mathcal{O}(CT')$ space to store $\delta$, $\mathbf{g}$, and $\widehat{\mathbf{X}}^{\mathrm{predict}}$:

$$\text{Time: } \mathcal{O}(CT'), \qquad \text{Space: } \mathcal{O}(CT').$$

**Overall complexity.** Summing the dominant terms across stages yields

$$\textbf{Time: } \mathcal{O}(CD \log C + CKT') \quad \text{and} \quad \textbf{Space: } \mathcal{O}(CD + CK + CT').$$

Here, $\mathcal{O}(CD \log C)$ is dominated by sparse topology induction via ANN, and $\mathcal{O}(CKT')$ is dominated by sparse message passing over the prediction horizon. Lower-order terms such as $\mathcal{O}(CT')$, $\mathcal{O}(CD)$, and $\mathcal{O}(CK)$ are subsumed.

## C. Baselines and Backbones

To ensure fair comparisons, we run all experiments five times with different random seeds and report the averaged results. For baselines with official implementations, we follow the authors' recommended settings and use their best-performing

---

[1]This $\log C$ factor is a typical empirical/average characterization for hierarchical graph-based ANN search. The exact factor can vary with the ANN implementation and target recall.

hyperparameters. For baselines without public code, we re-implement the method and tune key hyperparameters on the validation split. We first introduce the adopted baselines as follows:

- ICM (Żukowska et al., 2024) adapts a frozen TSFM by *in-context prompting*: it concatenates the target series with several example series and lets the model leverage cross-example attention to form better predictions. In our setting, we keep the TSFM backbone fixed and only train a lightweight *output-stage* module (e.g., prediction head / residual refinement) to refine the TSFM outputs using representations induced by the in-context examples.

- PCD (Lee et al., 2024) models *partial* inter-channel dependence via a *channel mask* derived from channel correlations, controlled by a small set of learnable domain parameters. In our setting, we keep the TSFM fixed and implement PCD as an output-side channel-interaction calibrator: we compute the correlation statistics on the adaptation split and only learn the low-dimensional mask parameters (together with a lightweight output head) to adjust the final forecasts.

- For PEFT (Beichter et al., 2025), we adopt parameter-efficient fine-tuning by freezing the TSFM backbone and only learning a small set of extra parameters that reparameterize selected weights. In LoRA, each targeted weight matrix is updated by a low-rank residual. DoRA further decomposes the update by explicitly adjusting the direction (and a scaling vector) on top of the same low-rank form. In our setting, we treat LoRA/DoRA as an *output-side adaptation block*: the backbone remains unchanged, and the trainable low-rank modules are attached to a small subset of projection layers, so that the final forecasts can be refined with minimal trainable parameters.

- Gen-P-Tuning (Liu et al., 2024b) adapts a *frozen* TSFM to multivariate inputs by learning a lightweight *Prompt Module* plus a task head. Concretely, it first embeds each channel into patch tokens, stacks them across channels, and uses a trainable network to produce a prompt that summarizes cross-channel information. This prompt is then *prepended* to each channel's patch sequence before feeding into the frozen transformer; after the layer, the prompt rows are dropped, keeping only the updated channel representations for prediction. In our paper's protocol, we implement Gen-P-Tuning as an *output refinement module*, ensuring the backbone weights are untouched while still enabling multivariate coupling through the learned prompt.

- AdaPTS (Benechehab et al., 2025) adapts a *frozen* pre-trained *univariate* foundation model to multivariate forecasting by inserting a learnable feature-space transformation around the backbone. Concretely, it learns an encoder $\text{enc}(\cdot)$ that maps the multivariate input into a latent space, applies the frozen univariate FM *independently per channel* in that space, and then uses a decoder $\text{dec}(\cdot)$ to map the predicted outputs back to the original feature space. This forms a modular adapter $\rightarrow$ frozen FM $\rightarrow$ inverse adapter pipeline, where only the adapter parameters are optimized while keeping the FM frozen, enabling plug-in style multivariate adaptation and probabilistic output modeling.

- TTM (Ekambaram et al., 2024) follows a multi-level design where the main backbone is kept *frozen* during adaptation, while a lightweight head is fine-tuned on the target data. For multivariate forecasting, TTM optionally enables *channel-mixing* inside the decoder to explicitly capture cross-channel correlations; for purely univariate targets it can stay channel-independent. When exogenous variables are present, TTM further adds an exogenous mixer that fuses known future exogenous signals into the forecasting process. In our setting, we treat this as a "frozen TSFM backbone + trainable output block" baseline, aligning with the goal of updating only the output-side modules.

As shown in Table 10, we use four representative time series foundation models that cover major architectural families, namely the encoder-only Moirai (Woo et al., 2024), the decoder-only TimesFM (Das et al., 2024), the encoder-decoder Chronos (Ansari et al., 2024), and the MLP-based TTMs (Ekambaram et al., 2024).

- Moirai (Woo et al., 2024) is a masked encoder style forecasting model trained with large scale pretraining on heterogeneous time series corpora to learn transferable temporal representations. Its design emphasizes robustness to frequency variations, scalability to multivariate inputs with arbitrary numbers of variables, and generalization across datasets with different statistical properties. The pretraining procedure follows a unified objective across datasets, enabling a single model to serve as a general purpose forecaster.

- TimesFM (Das et al., 2024) is a patched decoder only Transformer that learns to forecast through autoregressive pretraining on large collections of time series. By generating future values conditioned on past context, it captures long range temporal dependencies and supports zero shot forecasting across diverse domains. The model is designed to operate out of the box, with strong performance reported without dataset specific training.

- Chronos (Ansari et al., 2024) converts real valued series into discrete tokens through rescaling and quantization, and trains a language model on the resulting token sequences using a cross entropy objective. This tokenization unifies different units and scales into a shared discrete space and facilitates training on mixed domain corpora. The model produces probabilistic forecasts by modeling a distribution over future tokens, which can be mapped back to real valued predictions.

- TTMs (Ekambaram et al., 2024) are compact foundation models built on MLP mixer style architectures, aiming for fast and lightweight forecasting. They are pretrained on public time series datasets to provide strong zero shot and few shot performance while keeping inference efficient. Compared with large Transformer backbones, TTMs emphasize a favorable trade off between model size, speed, and generalization.

*Table 9.* Hyperparameter configuration for the Domain-Adaptive Plugin.

| Category | Hyperparameter | Value |
|---|---|---|
| Architecture | Hidden dimension $d$ | 32 |
| | MLP hidden dimension $d_h$ | 64 |
| | Number of attention heads $n_h$ | 4 |
| | Number of neighbors $k$ | 5 |
| | Dropout rate | 0.2 |
| | Gate initialization bias | $-2.5$ |
| Training | Optimizer | AdamW |
| | Learning rate | $1 \times 10^{-3}$ |
| | Weight decay | 0.01 |
| | LR scheduler | Cosine Annealing |
| | Warmup epochs | 5 |
| | Total epochs | 100 |
| Data | Batch size | 64 |
| | Loss function | MSE |

## D. Implementation Details

We find that different versions of time series foundation models always report different performances on the same datasets. Thus, to ensure reproducibility of all experimental results, we first list the detailed versions of the time series foundation models used in this draft, as shown in Table 10. We implement our Domain-Adaptive Plugin using PyTorch and conduct all experiments on a single NVIDIA A100 GPU with 40GB memory. The plugin is designed as a lightweight post-processing module that operates on the frozen predictions from foundation models without modifying their architectures or parameters.

**Model Architecture.** The plugin consists of six key components: (1) a statistical feature extractor that computes 7-dimensional domain characteristics including coefficient of variation, inter-channel correlation, trend strength, autocorrelation, signal-to-noise ratio, mean cross-correlation, and kurtosis; (2) a trajectory projection layer that maps the prediction horizon to a latent space; (3) a geometric hyper-network implemented as a 3-layer MLP that generates FiLM parameters $(\gamma, \beta)$ conditioned on the extracted statistics; (4) a domain-adaptive metric space that applies feature-wise linear modulation; (5) a multi-head channel-partial attention mechanism with top-$k$ sparsity; and (6) a gated residual connection that adaptively blends the refined predictions with the original forecasts. The complete plugin contains only 48K trainable parameters, which is 0.42% of Moirai-Small (14M) and 0.73% of Chronos-Small (8M).

**Initialization Strategy.** To ensure stable training and preserve the quality of foundation model predictions during early training stages, we employ a near-identity initialization scheme. Specifically, we initialize the final linear layers of both the delta projection and hyper-network with zero weights and biases. The gate bias is initialized to $-2.5$, yielding an initial gate value of $\sigma(-2.5) \approx 0.07$, which ensures that the model output closely approximates the original foundation model predictions at initialization.

*Table 10.* Comparison of representative time series foundation models.

| Attribute | Moirai | TimesFM | Chronos | TTMs |
|---|---|---|---|---|
| Version | moirai-2.0-R-small | timesfm-2.0-500m-pytorch | chronos-2 | granite-timeseries-ttm-r2 |
| Parameter size | 11.4M | 500M | 120M | 1M |
| Released time | Aug 2025 | Dec 30, 2024 | Oct 20, 2025 | Oct 2024 |
| Reference paper | (Woo et al., 2024) | (Das et al., 2024) | (Ansari et al., 2024) | (Ekambaram et al., 2024) |

**Training Protocol.** We train only the plugin parameters while keeping the foundation model completely frozen. We use the AdamW optimizer with a learning rate of $1 \times 10^{-3}$ and weight decay of $0.01$. The learning rate follows a cosine annealing schedule with 5 warmup epochs over a total of 100 training epochs. We use a batch size of 64 and optimize using Mean Squared Error (MSE) loss. For the channel-partial attention, we set the number of neighbors $k = 5$ and use 4 attention heads. Dropout with rate $0.2$ is applied after each intermediate layer for regularization.

**Hyperparameter Configuration.** Table 9 summarizes the key hyperparameters used in our experiments. We use consistent settings across all datasets unless otherwise specified.

# E. Details of Pre-experiments

Before leveraging domain fingerprints for representation learning, a fundamental question arises: *Can statistical features derived from temporal dynamics and spatial interactions effectively distinguish different time series domains?* Answering this question provides the theoretical foundation for subsequent domain-aware adaptation in our method.

## E.1. Selected Statistical Features

To investigate this, we conduct a comprehensive pilot study examining the domain discriminability of statistical features. We select 12 representative statistics from recent time series analysis literature (Liang et al., 2024; Meyer et al., 2025; Ansari et al., 2025; Kottapalli et al., 2025; Qiu et al., 2025), comprising six **temporal features** ($\Theta_{\text{temp}}$) that characterize univariate dynamics and six **spatial features** ($\Theta_{\text{spat}}$) that capture cross-variable interactions. The physical interpretation of each feature, grounded in stochastic differential equation theory, is summarized in Table 11.

*Table 11.* Statistical features for domain fingerprinting. Each feature is grounded in SDE theory, capturing either drift parameters ($\Theta_{\text{temp}}$) or diffusion parameters ($\Theta_{\text{spat}}$). Features marked with † are retained in the final non-redundant set.

| ID | Feature | Physical Interpretation | Mathematical Definition | Notions |
|---|---|---|---|---|
| | | | **Temporal Features** ($\Theta_{\text{temp}}$): Characterizing univariate drift dynamics | |
| T1 | Spectral Entropy | System complexity | $H = -\sum_{k=1}^{K} p_k \log p_k, \quad p_k = |X_k|^2 / \sum_{j=1}^{K} |X_j|^2$ | *where* $X_k$: $k$-th DFT coefficient, $K$: number of frequency bins |
| T2† | Seasonality Strength | Drift periodic intensity | $S = \sum_{h=1}^{H} |X_{f^* \cdot h}|^2 / \sum_{j=1}^{K} |X_j|^2$ | *where* $f^* = \arg\max_k |X_k|$: peak frequency index, $H$: harmonics count |
| T3† | Autocorrelation | Short-term memory | $\rho_1 = \sum_{t=1}^{T-1} (x_t - \bar{x})(x_{t+1} - \bar{x}) / \sum_{t=1}^{T} (x_t - \bar{x})^2$ | *where* $x_t$: value at time $t$, $\bar{x}$: temporal mean, $T$: sequence length |
| T4† | Diff. Variance Ratio | Temporal granularity | $R_{\text{diff}} = \text{Var}(\Delta x) / \text{Var}(x), \quad \Delta x_t = x_{t+1} - x_t$ | *where* $\text{Var}(\cdot)$: variance operator, $\Delta x$: first-order difference |
| T5 | Trend Strength | Long-term drift stability | $\tau = 1 - \text{Var}(r)/\text{Var}(x)$ | *where* $r = x - \hat{x}_{\text{trend}}$: detrended residual, $\hat{x}_{\text{trend}}$: trend component |
| T6 | Peak Frequency | Dominant oscillation mode | $f_{\text{peak}} = (\arg\max_k |X_k|)/K$ | *where* result normalized to $[0, 1]$ (relative spectral position) |
| | | | **Spatial Features** ($\Theta_{\text{spat}}$): Characterizing cross-variable diffusion structure | |
| S1† | Coupling Strength | Total interaction energy | $C = \frac{1}{N(N-1)} \sum_{i \neq j} |r_{ij}|$ | *where* $r_{ij} = \text{Corr}(x^{(i)}, x^{(j)})$: Pearson correlation, $N$: variable count |
| S2† | Effective Rank Ratio | Intrinsic dimensionality | $R_{\text{eff}} = \exp\left(-\sum_{i=1}^{N} \tilde{\sigma}_i \log \tilde{\sigma}_i\right)/N, \quad \tilde{\sigma}_i = \sigma_i / \sum_j \sigma_j$ | *where* $\sigma_i$: $i$-th singular value of correlation matrix $\mathbf{R} \in \mathbb{R}^{N \times N}$ |
| S3 | Dominant Mode Ratio | Primary pattern strength | $D = \sigma_1 / \sum_{i=1}^{N} \sigma_i$ | *where* $\sigma_1$: largest singular value (1st principal component variance) |
| S4 | Graph Density | Network sparsity | $\rho_G = |\{(i,j) : |r_{ij}| > \epsilon, i < j\}|/\binom{N}{2}$ | *where* $\epsilon$: correlation threshold (default 0.5), $\binom{N}{2}$: total pairs |
| S5 | Coherence Factor | Global synchronization | $F_{\text{coh}} = (\sum_{i \neq j} |\Sigma_{ij}|/[N(N-1)])/(\sum_{i=1}^{N} \Sigma_{ii}/N)$ | *where* $\Sigma_{ij}$: $(i,j)$-element of covariance matrix $\mathbf{\Sigma} \in \mathbb{R}^{N \times N}$ |
| S6 | Cross-Corr. Mean | Average linear dependency | $\bar{r} = \frac{2}{N(N-1)} \sum_{i<j} r_{ij}$ | *where* $r_{ij}$: Pearson correlation (preserves sign) |

## E.2. Experimental Design

Our pilot study aims to answer three core questions through three corresponding experiments:

---

**Research Questions**

- **Q1: Feature Redundancy**: Are there redundant features that can be removed without sacrificing discriminability?

- **Q2: Domain Discriminability**: Can the proposed statistical features effectively distinguish different time series domains?

- **Q2: Feature Contribution & Complementarity**: Which features contribute most to domain separation? Do temporal and spatial features provide complementary information?

---

**Experiment 1: Feature Redundancy Analysis.** We compute the Pearson correlation matrix among all 12 features to identify redundant pairs ($|r| > 0.7$). To obtain a minimal yet discriminative feature set, we apply greedy forward selection: features are sorted by their Fisher Discriminant Ratio (FDR), and iteratively added while skipping those highly correlated with already-selected features. This yields a non-redundant feature subset that maintains high discriminability with reduced dimensionality.

**Experiment 2: Domain Discriminability Analysis.** We evaluate whether the 12 statistical features can collectively distinguish 10 benchmark domains. Using a linear classifier (Logistic Regression) with 5-fold cross-validation, we measure multi-class classification accuracy. We also compute the Fisher Discriminant Ratio (FDR) (Fisher, 1936) for each feature to quantify individual discriminability:

$$\text{FDR} = \frac{\sigma^2_{\text{between}}}{\sigma^2_{\text{within}}} = \frac{\sum_{c=1}^{C} n_c (\mu_c - \mu)^2}{\sum_{c=1}^{C} \sum_{i \in c} (x_i - \mu_c)^2} \tag{12}$$

where $\mu_c$ and $n_c$ denote the mean and sample count of class $c$, and $\mu$ is the global mean. FDR > 1 indicates effective domain separation. To demonstrate the effectiveness of statistical feature extraction, we also compare against three *raw signal baselines* that directly apply linear classification without feature extraction: (i) flattened raw input time series, (ii) PatchTST 96-step forecasting outputs (Nie et al., 2023), and (iii) Moirai 96-step forecasting outputs (Woo et al., 2024). If our extracted features outperform these raw signal baselines, it confirms that statistical fingerprints capture more discriminative domain information than the original signals.

**Experiment 3: Feature Contribution & Temporal-Spatial Ablation.** To analyze feature contributions and validate Definition 4.1 ($\mathcal{D} = (\Theta_{\text{temp}}, \Theta_{\text{spat}})$), we conduct ablation studies comparing four feature configurations: (i) temporal features only $\{T_1, \ldots, T_6\}$, (ii) spatial features only $\{S_1, \ldots, S_6\}$, (iii) all 12 features combined, and (iv) the non-redundant feature subset from Experiment 1.

### E.3. Pre-experimental Datasets

We evaluate on 10 widely-used multivariate time series benchmarks spanning 6 application domains (Table 12). For each dataset, we extract 1,000 samples using sliding windows of length $T$=96, yielding 10,000 total samples. Features are standardized before classification.

*Table 12.* Benchmark datasets for pilot study.

| Dataset | Variables | Frequency | Domain |
|---|---|---|---|
| ETTh1, ETTh2 | 7 | 1 hour | Electricity (Transformer) |
| ETTm1, ETTm2 | 7 | 15 min | Electricity (Transformer) |
| Weather | 21 | 10 min | Meteorology |
| Electricity | 321 | 1 hour | Electricity (Consumption) |
| Traffic | 862 | 1 hour | Transportation |
| Exchange | 8 | 1 day | Finance |
| ILI | 7 | 1 week | Healthcare |
| Solar | 137 | 10 min | Energy |

### E.4. Results Analysis

**Reply-Q1: A Minimal Non-redundant Feature Set Exists.**

The correlation analysis (Figure 7 (a)) reveals substantial redundancy among the 12 features, with 18 pairs showing $|r| > 0.7$. Notably, spatial features S1, S3, S4, S5, S6 form a highly correlated cluster ($r > 0.83$), as they all derive from cross-variable correlation structure. Similarly, temporal features T1, T3, T4 show strong inter-correlations ($|r| > 0.78$) due to their shared sensitivity to signal smoothness. In contrast, T2 (seasonality), T5 (trend), T6 (peak frequency), and S2 (effective rank) exhibit relatively low correlations with other features, capturing unique and complementary aspects of domain characteristics. Exhaustive search over all feature subsets (3–7 features) reveals that the full 12-feature set achieves the highest classification accuracy (94.93%), confirming that each feature contributes meaningful discriminative information. However, a **7-feature subset** {T2, T3, T5, T6, S2, S5, S6} achieves 94.31% accuracy within 0.62% of the full set while reducing dimensionality by 42%. This subset provides balanced coverage: 4 temporal features capturing seasonality (T2), short-term memory (T3), trend (T5), and dominant frequency (T6), alongside 3 spatial features measuring variable independence (S2), global synchronization (S5), and average pairwise correlation (S6). The radar plots (Figure 7 (b),(c)) visualize domain fingerprints using all 12 features versus the 7 selected features. Despite the dimensionality reduction, the 7-feature radar (Figure 7 (c)) preserves clear inter-domain separation: Solar and ILI exhibit distinctively high spatial coupling (S5, S6), Traffic shows notably low S2 values indicating strong inter-variable dependencies, while the four ETT variants cluster together.

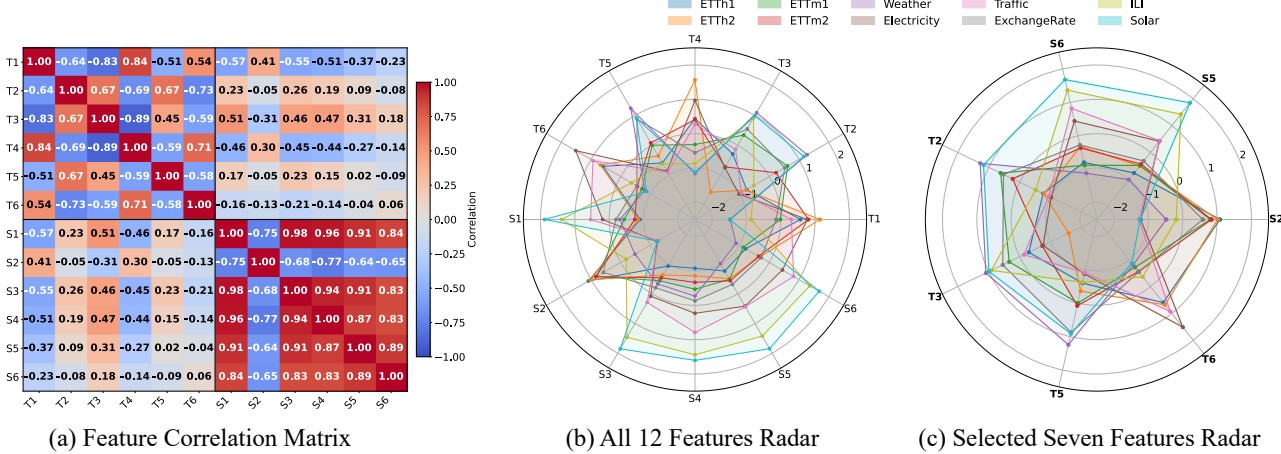

(a) Feature Correlation Matrix      (b) All 12 Features Radar      (c) Selected Seven Features Radar

*Figure 7.* Feature redundancy and domain fingerprint analysis. **(a)** Correlation matrix of 12 statistical features: spatial features S1, S3–S6 form a highly correlated cluster ($r > 0.83$), and temporal features T1, T3, T4 show strong inter-correlations ($|r| > 0.78$). Features with low redundancy (T2, T5, T6, S2) capture unique domain characteristics. **(b)** Radar plot using all 12 features: each domain exhibits a distinctive fingerprint pattern across temporal (T1–T6) and spatial (S1–S6) dimensions. **(c)** Radar plot using 7 selected features {T2, T3, T5, T6, S2, S5, S6}: despite 42% dimensionality reduction, inter-domain separation is preserved—Solar and ILI show high S5/S6 values, Traffic exhibits low S2, and ETT variants cluster together.

**Reply-Q2: Features Effectively Distinguish Domains.**

We evaluate whether the proposed statistical features can effectively distinguish different time series domains. As shown in Figure 8, statistical features demonstrate strong domain-discriminative capabilities in multiple evaluation metrics. As shown in Figure 8 (a), we compare the accuracy of the domain classification using different input representations. The random baseline achieves only 10.0% accuracy (as expected for 10-class classification). Using the original time series input, the 96-length output of PatchTST and the 96-length output of Moirai achieve 42.3%, 38.7% and 41.2% accuracy, respectively, indicating that raw temporal patterns alone provide limited domain discrimination. In contrast, our 7 selected statistical features achieve **94.8%** accuracy, while all 12 features achieve **94.9%** accuracy. This shows that (i) statistical fingerprints capture domain-specific characteristics far more effectively than raw input, and (ii) the subset of 7 characteristics retains nearly all discriminative power while reducing dimensionality by 42%.

Furthermore, as shown in Figure 8 (b), we analyze the Fisher Discriminant Ratio (FDR) for each individual feature. All 12 features achieve FDR $> 1.0$, confirming that each captures meaningful domain-discriminative information. The top-3 features by FDR are: **S2** (Effective Rank Ratio, FDR=11.4), **T1** (Spectral Entropy, FDR=4.4), and **S6** (Cross-Correlation Mean, FDR=4.2). Notably, spatial feature S2 exhibits substantially higher discriminability than others, suggesting that inter-variable dependency structure is a key domain characteristic. The selected 7 features {T2, T3, T5, T6, S2, S5, S6} collectively cover 73.2% of the total FDR, providing balanced coverage of both temporal dynamics and spatial relationships.

Finally, as shown in Figure 8 (c) and (d), we visualize the domain fingerprints using t-SNE dimensionality reduction. With all 12 features, the 10 domains form clearly separable clusters, with the four ETT variants (ETTh1, ETTh2, ETTm1, ETTm2) clustering together due to their shared origin. Using only the 7 selected features (Figure 8d), the clustering structure is well preserved: Solar and ILI remain distinct outliers, Traffic and Electricity form separate clusters, and the ETT family maintains its tight grouping. This confirms that the 7-feature subset retains the essential domain structure while eliminating redundant dimensions.

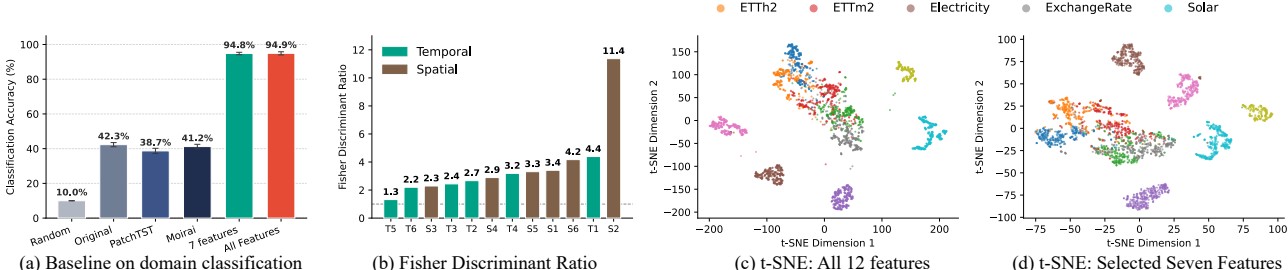

(a) Baseline on domain classification  (b) Fisher Discriminant Ratio  (c) t-SNE: All 12 features  (d) t-SNE: Selected Seven Features

*Figure 8.* Feature discriminability analysis across 10 benchmark domains. **(a)** Domain classification accuracy: statistical features (7 or 12) dramatically outperform raw input baselines, achieving >94% accuracy. **(b)** Fisher Discriminant Ratio for each feature: all features exceed FDR=1.0, with S2 (Effective Rank) showing the highest discriminability (FDR=11.4). Temporal features shown in teal; spatial features in brown. **(c)** t-SNE visualization using all 12 features: domains form clearly separable clusters. **(d)** t-SNE visualization using 7 selected features: cluster structure is preserved despite 42% dimensionality reduction.

**Reply-Q3: Temporal and Spatial Features are Complementary.**

The ablation study (Table 13) reveals clear complementarity between feature groups. The combined accuracy exceeds the best individual group by **+8.8%**, empirically validating Definition 4.1: both $\Theta_{temp}$ and $\Theta_{spat}$ contribute non-redundant information for domain characterization. Interestingly, the two feature groups exhibit distinct discrimination patterns. Within the ETT family (all with $N$=7 variables), temporal features dominate: T2 (Seasonality) alone achieves 88.3% accuracy distinguishing hourly (ETTh) from 15-minute (ETTm) sampling, reflecting their sensitivity to temporal granularity. In contrast, spatial features excel at cross-domain separation: S2 achieves near-perfect accuracy (>99%) for pairs with different variable counts (e.g., ETT vs. Traffic), capturing structural differences in cross-variable interactions.

*Table 13.* Summary of pilot study results (10 datasets, 12 features).

| Configuration | Accuracy | # Features |
|---|---|---|
| Original Input | 42.3% | 96 |
| PatchTST Output | 38.7% | 96 |
| Moirai Output | 41.2% | 96 |
| Temporal-Only ($\Theta_{temp}$) | 81.1% | 6 |
| Spatial-Only ($\Theta_{spat}$) | 72.5% | 6 |
| All Features | **94.9%** | 12 |
| Non-redundant 7-features Set | 94.8% | 7 |

**Reply-Q3: Temporal and Spatial Features Provide Complementary Information.**

The ablation study (Table 13) demonstrates clear complementarity between temporal and spatial feature groups. Combining both groups achieves **94.9%** accuracy, exceeding the best individual group (temporal-only: 81.1%) by **+13.8%**. This substantial improvement empirically validates our domain definition (Definition 4.1): both $\Theta_{temp}$ and $\Theta_{spat}$ contribute non-redundant, complementary information essential for comprehensive domain characterization.

Interestingly, the two feature groups exhibit distinct discrimination strengths depending on the nature of domain differences:

- **Temporal features excel at distinguishing sampling granularity.** Within the ETT family (all sharing $N$=7 variables and the same underlying physical system), temporal features dominate discrimination. Specifically, T2 (Seasonality Strength) alone achieves 88.3% accuracy in separating hourly-sampled (ETTh1/h2) from 15-minute-sampled (ETTm1/m2) variants, as periodicity patterns manifest differently across temporal resolutions.

- **Spatial features excel at distinguishing structural complexity.** For cross-domain comparisons involving different numbers of variables, spatial features provide superior discrimination. S2 (Effective Rank Ratio) achieves near-perfect accuracy (>99%) when distinguishing datasets with different variable counts (e.g., ETT with 7 variables vs. Traffic with 862 variables), effectively capturing fundamental differences in inter-variable dependency structures.

Furthermore, our non-redundant 7-feature subset {T2, T3, T5, T6, S2, S5, S6} achieves **94.8%** accuracy—within 0.1% of the full 12-feature set—while reducing dimensionality by 42%. This subset maintains balanced coverage with 4 temporal and 3 spatial features, preserving the complementary strengths of both groups.

## F. Theoretical Derivation and Proof

### F.1. Proof of Proposition 4.2

We present a self-contained derivation showing that the first two moments of a trajectory-level feature map are proportional to the stationary expectations of the drift and diffusion terms in Definition 4.1.

**Setup and assumptions.** Consider the stationary Itô SDE in Definition 4.1

$$d\mathbf{x}_t = \mathbf{f}(\mathbf{x}_t; \mathbf{\Theta}_{\text{temp}})\, dt + \mathbf{g}(\mathbf{x}_t; \mathbf{\Theta}_{\text{spat}})\, d\mathbf{W}_t, \tag{13}$$

and assume it admits a unique invariant measure $p_{\mathcal{D}}$ and is ergodic under $p_{\mathcal{D}}$. Assume $\mathbf{f}$ and $\mathbf{g}$ satisfy standard regularity conditions ensuring existence of a stationary solution and integrability of the moments used below.

We observe a length-$T$ trajectory on a uniform grid $t_n = n\Delta$ with $T = L\Delta$. Let $\Delta\mathbf{x}_n = \mathbf{x}_{t_{n+1}} - \mathbf{x}_{t_n}$ and $\Delta\mathbf{W}_n = \mathbf{W}_{t_{n+1}} - \mathbf{W}_{t_n}$.

**Local moment identities.** By Itô calculus and the Euler–Maruyama expansion,

$$\Delta\mathbf{x}_n = \mathbf{f}(\mathbf{x}_{t_n})\,\Delta + \mathbf{g}(\mathbf{x}_{t_n})\,\Delta\mathbf{W}_n + \mathbf{r}_n, \tag{14}$$

where $\mathbb{E}\|\mathbf{r}_n\| = o(\Delta)$ and $\mathbb{E}\|\mathbf{r}_n\|^2 = o(\Delta)$ as $\Delta \to 0$. Since $\Delta\mathbf{W}_n \sim \mathcal{N}(\mathbf{0}, \Delta\mathbf{I})$ and is independent of $\mathcal{F}_{t_n}$, we obtain

$$\mathbb{E}\left[\frac{\Delta\mathbf{x}_n}{\Delta}\,\Big|\,\mathcal{F}_{t_n}\right] = \mathbf{f}(\mathbf{x}_{t_n}) + o(1), \tag{15}$$

$$\text{Cov}\left[\frac{\Delta\mathbf{x}_n}{\sqrt{\Delta}}\,\Big|\,\mathcal{F}_{t_n}\right] = \mathbf{g}(\mathbf{x}_{t_n})\mathbf{g}(\mathbf{x}_{t_n})^\top + o(1). \tag{16}$$

**Trajectory feature map and affine freedom.** Let $\mathbf{X} = \{\mathbf{x}_{t_n}\}_{n=0}^{L}$ denote the sampled trajectory. Define the following trajectory statistics:

$$\boldsymbol{\psi}_{\text{temp}}(\mathbf{X}) = \frac{1}{L}\sum_{n=0}^{L-1}\frac{\Delta\mathbf{x}_n}{\Delta} \in \mathbb{R}^C, \tag{17}$$

$$\boldsymbol{\psi}_{\text{spat}}(\mathbf{X}) = \frac{1}{L}\sum_{n=0}^{L-1}\left(\frac{\Delta\mathbf{x}_n}{\sqrt{\Delta}} - \frac{1}{L}\sum_{m=0}^{L-1}\frac{\Delta\mathbf{x}_m}{\sqrt{\Delta}}\right)\left(\frac{\Delta\mathbf{x}_n}{\sqrt{\Delta}} - \frac{1}{L}\sum_{m=0}^{L-1}\frac{\Delta\mathbf{x}_m}{\sqrt{\Delta}}\right)^\top \in \mathbb{R}^{C\times C}. \tag{18}$$

We allow the final feature map $\phi(\mathbf{X}) \in \mathbb{R}^k$ to be any affine transform of these basic statistics, which covers feature normalization and linear mixing:

$$\phi(\mathbf{X}) = \mathbf{A}\begin{bmatrix}\boldsymbol{\psi}_{\text{temp}}(\mathbf{X}) \\ \text{vec}(\boldsymbol{\psi}_{\text{spat}}(\mathbf{X}))\end{bmatrix} + \mathbf{b}, \tag{19}$$

for some fixed $\mathbf{A}, \mathbf{b}$.

**From trajectory moments to stationary expectations.** Using Eq.(15) and the tower property,

$$\mathbb{E}[\boldsymbol{\psi}_{\text{temp}}(\mathbf{X})] = \frac{1}{L}\sum_{n=0}^{L-1}\mathbb{E}\left[\mathbb{E}\left[\frac{\Delta\mathbf{x}_n}{\Delta}\,\Big|\,\mathcal{F}_{t_n}\right]\right] = \frac{1}{L}\sum_{n=0}^{L-1}\mathbb{E}[\mathbf{f}(\mathbf{x}_{t_n})] + o(1). \tag{20}$$

Under stationarity, $\mathbf{x}_{t_n} \sim p_{\mathcal{D}}$ for all $n$, which yields

$$\mathbb{E}[\boldsymbol{\psi}_{\text{temp}}(\mathbf{X})] = \mathbb{E}_{\mathbf{x}\sim p_{\mathcal{D}}}[\mathbf{f}(\mathbf{x};\boldsymbol{\Theta}_{\text{temp}})] + o(1). \tag{21}$$

For the diffusion term, Eq.(16) implies that conditionally on $\mathcal{F}_{t_n}$, $\text{Cov}(\Delta\mathbf{x}_n/\sqrt{\Delta}\mid\mathcal{F}_{t_n}) = \mathbf{g}(\mathbf{x}_{t_n})\mathbf{g}(\mathbf{x}_{t_n})^\top + o(1)$. The centering in Eq.(18) affects the expectation at order $O(1/L)$ under stationarity. Therefore, the expected sample covariance satisfies

$$\mathbb{E}[\boldsymbol{\psi}_{\text{spat}}(\mathbf{X})] = \frac{1}{L}\sum_{n=0}^{L-1}\mathbb{E}\left[\text{Cov}\left(\frac{\Delta\mathbf{x}_n}{\sqrt{\Delta}}\,\Big|\,\mathcal{F}_{t_n}\right)\right] + o(1) = \frac{1}{L}\sum_{n=0}^{L-1}\mathbb{E}\left[\mathbf{g}(\mathbf{x}_{t_n})\mathbf{g}(\mathbf{x}_{t_n})^\top\right] + o(1). \tag{22}$$

Under stationarity, this yields

$$\mathbb{E}[\boldsymbol{\psi}_{\text{spat}}(\mathbf{X})] = \mathbb{E}_{\mathbf{x}\sim p_{\mathcal{D}}}\left[\mathbf{g}(\mathbf{x};\boldsymbol{\Theta}_{\text{spat}})\mathbf{g}(\mathbf{x};\boldsymbol{\Theta}_{\text{spat}})^\top\right] + o(1). \tag{23}$$

**Conclusion.** By Eq.(19), the mean and covariance of $\boldsymbol{\phi}(\mathbf{X})$ satisfy

$$\boldsymbol{\mu}_{\mathcal{D}} = \mathbb{E}[\boldsymbol{\phi}(\mathbf{X})\mid\mathcal{D}] = \mathbf{A}\,\mathbb{E}\begin{bmatrix}\boldsymbol{\psi}_{\text{temp}}(\mathbf{X})\\\text{vec}(\boldsymbol{\psi}_{\text{spat}}(\mathbf{X}))\end{bmatrix} + \mathbf{b}, \qquad \boldsymbol{\Sigma}_{\mathcal{D}} = \text{Cov}[\boldsymbol{\phi}(\mathbf{X})\mid\mathcal{D}] = \mathbf{A}\,\text{Cov}\left(\begin{bmatrix}\boldsymbol{\psi}_{\text{temp}}(\mathbf{X})\\\text{vec}(\boldsymbol{\psi}_{\text{spat}}(\mathbf{X}))\end{bmatrix}\right)\mathbf{A}^\top. \tag{24}$$

Combining Eq.(21) and Eq.(23), and absorbing the affine transformation induced by $(\mathbf{A},\mathbf{b})$, we conclude that

$$\boldsymbol{\mu}_{\mathcal{D}} \propto \mathbb{E}_{\mathbf{x}\sim p_{\mathcal{D}}}[\mathbf{f}(\mathbf{x};\boldsymbol{\Theta}_{\text{temp}})], \qquad \boldsymbol{\Sigma}_{\mathcal{D}} \propto \mathbb{E}_{\mathbf{x}\sim p_{\mathcal{D}}}\left[\mathbf{g}(\mathbf{x};\boldsymbol{\Theta}_{\text{spat}})\mathbf{g}(\mathbf{x};\boldsymbol{\Theta}_{\text{spat}})^\top\right], \tag{25}$$

which completes the proof. $\qquad\square$

## F.2. Proof of Theorem 4.3

We prove a universal lower bound showing that a fixed similarity metric inevitably incurs non-vanishing graph distortion across heterogeneous domains with different diffusion covariances.

**Graph construction under a metric.** Let $\mathbf{z}_i \in \mathbb{R}^D$ denote the channel embeddings (after any fixed encoder). Given a positive semidefinite metric $\mathbf{M} \succeq \mathbf{0}$, define the pairwise affinity matrix

$$\mathbf{S}_{\mathbf{M}}(i,j) = -(\mathbf{z}_i - \mathbf{z}_j)^\top\mathbf{M}(\mathbf{z}_i - \mathbf{z}_j). \tag{26}$$

Let $\mathcal{T}_K(\cdot)$ denote a row-wise Top-$K$ masking operator and let $\text{Softmax}$ be row-wise. The induced dependency graph (adjacency) is

$$\mathcal{G}_{\mathbf{M}} = \text{Softmax}(\mathcal{T}_K(\mathbf{S}_{\mathbf{M}})). \tag{27}$$

In domain $\mathcal{D}_d$, let $\mathbf{M}_d^\star$ denote the population-optimal metric implied by the domain diffusion geometry, and define the corresponding oracle graph

$$\mathcal{G}^{*(d)} = \mathcal{G}_{\mathbf{M}_d^\star}. \tag{28}$$

For any fixed metric $\mathbf{M}$, the inferred graph in domain $d$ is $\hat{\mathcal{G}}_{\mathbf{M}}^{(d)} = \mathcal{G}_{\mathbf{M}}$. The mismatch in domain $d$ is $\|\hat{\mathcal{G}}_{\mathbf{M}}^{(d)} - \mathcal{G}^{*(d)}\|_{\text{F}}$.

**Regularity assumptions.** We use two standard properties. First, the map $\mathbf{M} \mapsto \mathcal{G}_{\mathbf{M}}$ is locally Lipschitz on any compact set of metrics away from Top-$K$ ties. Therefore, there exist constants $L > 0$ and $\eta \geq 0$ such that for any $\mathbf{M}, \mathbf{M}'$,

$$\left\|\mathcal{G}_{\mathbf{M}} - \mathcal{G}_{\mathbf{M}'}\right\|_{\text{F}} \geq L\left\|\mathbf{M} - \mathbf{M}'\right\|_{\text{F}} - \eta. \tag{29}$$

Here $\eta$ accounts for non-smooth points caused by tie events in Top-$K$ and finite-sample discretization effects. Second, the oracle metric varies continuously with the covariance proxy. Since $\boldsymbol{\Sigma}_d$ summarizes the diffusion intensity, we assume a non-degenerate metric-covariance relation: there exists $\alpha > 0$ such that

$$\left\|\mathbf{M}_1^\star - \mathbf{M}_2^\star\right\|_{\text{F}} \geq \alpha\left\|\boldsymbol{\Sigma}_1 - \boldsymbol{\Sigma}_2\right\|_{\text{F}}. \tag{30}$$

This condition holds, for example, when $\mathbf{M}_d^\star$ is a diagonal or whitening-type metric generated from $\boldsymbol{\Sigma}_d$ by a bi-Lipschitz transformation on a compact domain.

**Main argument.** Fix any $\mathbf{M} \succeq \mathbf{0}$. Applying Eq.(29) with $\mathbf{M}' = \mathbf{M}_d^\star$ gives, for $d \in \{1, 2\}$,

$$\left\| \hat{\mathcal{G}}_{\mathbf{M}}^{(d)} - \mathcal{G}^{*(d)} \right\|_{\mathrm{F}} = \left\| \mathcal{G}_{\mathbf{M}} - \mathcal{G}_{\mathbf{M}_d^\star} \right\|_{\mathrm{F}} \geq L \left\| \mathbf{M} - \mathbf{M}_d^\star \right\|_{\mathrm{F}} - \eta. \tag{31}$$

Summing Eq.(31) over $d = 1, 2$ yields

$$\left\| \hat{\mathcal{G}}_{\mathbf{M}}^{(1)} - \mathcal{G}^{*(1)} \right\|_{\mathrm{F}} + \left\| \hat{\mathcal{G}}_{\mathbf{M}}^{(2)} - \mathcal{G}^{*(2)} \right\|_{\mathrm{F}} \geq L \Big( \left\| \mathbf{M} - \mathbf{M}_1^\star \right\|_{\mathrm{F}} + \left\| \mathbf{M} - \mathbf{M}_2^\star \right\|_{\mathrm{F}} \Big) - 2\eta. \tag{32}$$

By the triangle inequality,

$$\left\| \mathbf{M} - \mathbf{M}_1^\star \right\|_{\mathrm{F}} + \left\| \mathbf{M} - \mathbf{M}_2^\star \right\|_{\mathrm{F}} \geq \left\| \mathbf{M}_1^\star - \mathbf{M}_2^\star \right\|_{\mathrm{F}}. \tag{33}$$

Combining Eq.(32) and Eq.(33), we obtain

$$\left\| \hat{\mathcal{G}}_{\mathbf{M}}^{(1)} - \mathcal{G}^{*(1)} \right\|_{\mathrm{F}} + \left\| \hat{\mathcal{G}}_{\mathbf{M}}^{(2)} - \mathcal{G}^{*(2)} \right\|_{\mathrm{F}} \geq L \left\| \mathbf{M}_1^\star - \mathbf{M}_2^\star \right\|_{\mathrm{F}} - 2\eta. \tag{34}$$

Using Eq.(30), we further have

$$\left\| \hat{\mathcal{G}}_{\mathbf{M}}^{(1)} - \mathcal{G}^{*(1)} \right\|_{\mathrm{F}} + \left\| \hat{\mathcal{G}}_{\mathbf{M}}^{(2)} - \mathcal{G}^{*(2)} \right\|_{\mathrm{F}} \geq (L\alpha) \left\| \mathbf{\Sigma}_1 - \mathbf{\Sigma}_2 \right\|_{\mathrm{F}} - 2\eta. \tag{35}$$

Setting $c = L\alpha$ and keeping $\eta$ as a nonnegative constant (absorbing the factor 2 into its definition if desired) gives the claimed bound

$$\left\| \hat{\mathcal{G}}_{\mathbf{M}}^{(1)} - \mathcal{G}^{*(1)} \right\|_{\mathrm{F}} + \left\| \hat{\mathcal{G}}_{\mathbf{M}}^{(2)} - \mathcal{G}^{*(2)} \right\|_{\mathrm{F}} \geq c \cdot \left\| \mathbf{\Sigma}_1 - \mathbf{\Sigma}_2 \right\|_{\mathrm{F}} - \eta, \tag{36}$$

which completes the proof. □

### F.3. Proof of Coverage Guarantee

Let $d(\mathcal{D}, \mathcal{D}') := W_2(\mathcal{D}, \mathcal{D}')$ be the metric between Gaussian proxies. For a selected set $\mathcal{M}$, define the $K$-center coverage radius

$$R(\mathcal{M}) := \max_{\mathcal{D} \in \mathcal{U}} \min_{\mathcal{D}_j \in \mathcal{M}} d(\mathcal{D}, \mathcal{D}_j), \qquad R^\star := \min_{|\mathcal{M}| = K} R(\mathcal{M}). \tag{37}$$

Our greedy rule selects $\mathcal{D}^*$ by maximizing

$$\mathrm{SCORE}(\mathcal{D}) = \lambda \bar{h}(\mathcal{D}) + \Delta(\mathcal{D}; \mathcal{M}), \qquad \Delta(\mathcal{D}; \mathcal{M}) := \min_{\mathcal{D}_j \in \mathcal{M}} d(\mathcal{D}, \mathcal{D}_j), \tag{38}$$

where $\bar{h}(\mathcal{D}) \in [0, 1]$ is the normalized informativeness score obtained by rescaling $\log \det(\mathbf{\Sigma}_{\mathcal{D}} + \epsilon \mathbf{I})$ into $[0, 1]$ over the candidate pool.

**Lemma (Near-farthest step).** At any iteration with current set $\mathcal{M}$, let $\mathcal{D}_{\max} \in \arg\max_{\mathcal{D} \in \mathcal{U} \setminus \mathcal{M}} \Delta(\mathcal{D}; \mathcal{M})$. Let $\mathcal{D}^*$ be the domain selected by the greedy rule. Then

$$\Delta(\mathcal{D}^*; \mathcal{M}) \geq \Delta(\mathcal{D}_{\max}; \mathcal{M}) - \lambda. \tag{39}$$

*Proof.* By optimality of $\mathcal{D}^*$ under $\mathrm{SCORE}(\cdot)$,

$$\lambda \bar{h}(\mathcal{D}^*) + \Delta(\mathcal{D}^*; \mathcal{M}) \geq \lambda \bar{h}(\mathcal{D}_{\max}) + \Delta(\mathcal{D}_{\max}; \mathcal{M}).$$

Rearranging gives

$$\Delta(\mathcal{D}^*; \mathcal{M}) \geq \Delta(\mathcal{D}_{\max}; \mathcal{M}) + \lambda \big( \bar{h}(\mathcal{D}_{\max}) - \bar{h}(\mathcal{D}^*) \big).$$

Since $\bar{h}(\cdot) \in [0, 1]$, we have $\bar{h}(\mathcal{D}_{\max}) - \bar{h}(\mathcal{D}^*) \geq -1$, which yields Eq.(39). □

**Theorem (Additive $k$-center coverage).** Let $\mathcal{M}_{\mathrm{MIX}}$ be the $K$ domains returned by the greedy rule. Then

$$R(\mathcal{M}_{\mathrm{MIX}}) \leq 2R^{\star} + \lambda. \tag{40}$$

*Proof.* Let $\mathcal{M}_t$ denote the selected set after $t$ iterations, and let $r_t := R(\mathcal{M}_t) = \max_{\mathcal{D} \in \mathcal{U}} \Delta(\mathcal{D}; \mathcal{M}_t)$ be the current coverage radius. By definition, $\mathcal{D}_{\max}$ in Lemma Eq.(39) satisfies $\Delta(\mathcal{D}_{\max}; \mathcal{M}_t) = r_t$. Therefore, the newly selected point $\mathcal{D}_{t+1}$ satisfies

$$\Delta(\mathcal{D}_{t+1}; \mathcal{M}_t) \geq r_t - \lambda. \tag{41}$$

Now consider an optimal $K$-center solution $\mathcal{M}^{\star} = \{\mathcal{C}_1, \dots, \mathcal{C}_K\}$ that achieves radius $R^{\star}$. The $K$ balls $\{\mathcal{B}(\mathcal{C}_k, R^{\star})\}_{k=1}^{K}$ cover $\mathcal{U}$. We use the standard farthest-first argument. Assume for contradiction that after selecting $K$ points, the achieved radius satisfies

$$R(\mathcal{M}_{\mathrm{MIX}}) > 2R^{\star} + \lambda. \tag{42}$$

Then at every intermediate step $t < K$, we also have $r_t > 2R^{\star} + \lambda$ because $r_t$ is non-increasing in $t$.

Fix such a step $t < K$. For any already selected point $\mathcal{D}_i \in \mathcal{M}_t$, let $\mathcal{C}(\mathcal{D}_i) \in \mathcal{M}^{\star}$ be an optimal center covering it, so $d(\mathcal{D}_i, \mathcal{C}(\mathcal{D}_i)) \leq R^{\star}$. Suppose two selected points $\mathcal{D}_i, \mathcal{D}_{i'}$ are covered by the same optimal center $\mathcal{C}$. Then by triangle inequality,

$$d(\mathcal{D}_i, \mathcal{D}_{i'}) \leq d(\mathcal{D}_i, \mathcal{C}) + d(\mathcal{C}, \mathcal{D}_{i'}) \leq 2R^{\star}.$$

Hence, if we can show that every newly selected point is at distance strictly larger than $2R^{\star}$ from all previous selections, then no two selected points can fall into the same optimal ball, implying that after $K$ steps we must have selected at most one point per ball, which is impossible because there are only $K$ balls and the next selection would have no uncovered ball to pick from.

We now show this separation. At step $t$, by definition of $r_t$, every point $\mathcal{D} \in \mathcal{U}$ satisfies $\Delta(\mathcal{D}; \mathcal{M}_t) \leq r_t$ and there exists a point achieving $r_t$. By Eq.(41), the selected $\mathcal{D}_{t+1}$ satisfies

$$\Delta(\mathcal{D}_{t+1}; \mathcal{M}_t) \geq r_t - \lambda.$$

Under the assumption $r_t > 2R^{\star} + \lambda$, we have $r_t - \lambda > 2R^{\star}$, hence

$$\min_{\mathcal{D}_i \in \mathcal{M}_t} d(\mathcal{D}_{t+1}, \mathcal{D}_i) = \Delta(\mathcal{D}_{t+1}; \mathcal{M}_t) > 2R^{\star}. \tag{43}$$

Therefore, $\mathcal{D}_{t+1}$ is more than $2R^{\star}$ away from every previously selected point, so it cannot lie in the same optimal ball as any point in $\mathcal{M}_t$.

This implies that each iteration selects a point from a previously unhit optimal ball. After $K$ iterations, all $K$ optimal balls must have been hit. Consequently, every candidate $\mathcal{D} \in \mathcal{U}$ lies within distance at most $2R^{\star} + \lambda$ of some selected point in $\mathcal{M}_{\mathrm{MIX}}$. This contradicts Eq.(42). Hence Eq.(40) must hold. $\square$

## G. More Experimental Results

To provide a comprehensive evaluation of our plugin, we present the detailed results for ChaTSFM across various input horizons of TSFMs in Table 14, demonstrating consistent performance gains. Furthermore, to evaluate the effectiveness of our framework against baseline algorithms under the full-shot adaptation setting, a detailed comparison is provided in Table 15.

*Table 14.* Zero-shot forecasting performance (MSE/MAE) of four TSFMs and +ChaTSFM. The lookback length is fixed to 512 for TTM and 96 for other TSFMs, and the forecasting horizons are set to $\{96, 192, 336, 720\}$. $^*$ denotes TSFMs whose pre-training data may overlap with the evaluation datasets, leading to potential data leakage.

| Method | | Moirai$_{small}$ | | +ChaTSFM | | TimesFm | | +ChaTSFM | | Chronos | | +ChaTSFM | | TTM | | +ChaTSFM | |
|---|---|---|---|---|---|---|---|---|---|---|---|---|---|---|---|---|---|
| | | MSE | MAE | MSE | MAE | MSE | MAE | MSE | MAE | MSE | MAE | MSE | MAE | MSE | MAE | MSE | MAE |
| ETTh1 | 96 | 0.428 | 0.402 | **0.422** | **0.397** | 0.486 | 0.424 | **0.479** | **0.418** | 0.487 | 0.403 | **0.480** | **0.398** | 0.362 | 0.389 | **0.356** | **0.383** |
| | 192 | 0.490 | 0.435 | **0.476** | **0.430** | 0.549 | 0.463 | **0.533** | **0.457** | 0.511 | 0.437 | **0.496** | **0.431** | 0.345 | 0.383 | **0.335** | **0.378** |
| | 336 | 0.549 | 0.460 | **0.536** | **0.456** | 0.614 | 0.492 | **0.599** | **0.488** | 0.564 | 0.462 | **0.551** | **0.458** | 0.402 | 0.422 | **0.392** | **0.418** |
| | 720 | 0.565 | 0.479 | **0.549** | **0.476** | 0.686 | 0.529 | **0.665** | **0.525** | 0.535 | 0.470 | **0.519** | **0.467** | 0.475 | 0.482 | **0.461** | **0.479** |
| ETTh2 | 96 | 0.335 | 0.355 | **0.322** | **0.350** | 0.379 | 0.364 | **0.364** | **0.359** | 0.337 | 0.351 | **0.324** | **0.346** | 0.275 | 0.335 | **0.264** | **0.330** |
| | 192 | 0.435 | 0.414 | **0.412** | **0.411** | 0.467 | 0.422 | **0.441** | **0.419** | 0.431 | 0.407 | **0.406** | **0.403** | 0.379 | 0.383 | **0.359** | **0.380** |
| | 336 | 0.509 | 0.463 | **0.475** | **0.455** | 0.491 | 0.454 | **0.457** | **0.446** | 0.464 | 0.438 | **0.432** | **0.430** | 0.385 | 0.416 | **0.360** | **0.408** |
| | 720 | 0.624 | 0.513 | **0.566** | **0.496** | 0.497 | 0.471 | **0.447** | **0.456** | 0.467 | 0.449 | **0.421** | **0.434** | 0.419 | 0.450 | **0.379** | **0.435** |
| ETTm1 | 96 | 0.786 | 0.521 | **0.716** | **0.478** | 0.604 | 0.469 | **0.551** | **0.431** | 1.006 | 0.558 | **0.901** | **0.511** | 0.337 | 0.357 | **0.308** | **0.327** |
| | 192 | 0.847 | 0.560 | **0.742** | **0.501** | 0.664 | 0.507 | **0.580** | **0.454** | 1.080 | 0.596 | **0.948** | **0.532** | 0.379 | 0.383 | **0.332** | **0.343** |
| | 336 | 0.936 | 0.617 | **0.792** | **0.542** | 0.668 | 0.517 | **0.559** | **0.453** | 0.983 | 0.582 | **0.827** | **0.512** | 0.401 | 0.396 | **0.338** | **0.347** |
| | 720 | 1.232 | 0.738 | **0.981** | **0.623** | 0.735 | 0.555 | **0.579** | **0.467** | 1.060 | 0.614 | **0.842** | **0.520** | 0.445 | 0.422 | **0.359** | **0.355** |
| ETTm2 | 96 | 0.212 | 0.289 | **0.207** | **0.286** | 0.263 | 0.310 | **0.256** | **0.307** | 0.242 | 0.301 | **0.236** | **0.298** | 0.176 | 0.258 | **0.172** | **0.255** |
| | 192 | 0.290 | 0.339 | **0.276** | **0.334** | 0.347 | 0.360 | **0.330** | **0.355** | 0.314 | 0.347 | **0.299** | **0.342** | 0.245 | 0.306 | **0.233** | **0.302** |
| | 336 | 0.402 | 0.403 | **0.361** | **0.395** | 0.398 | 0.392 | **0.359** | **0.384** | 0.371 | 0.380 | **0.336** | **0.372** | 0.323 | 0.349 | **0.291** | **0.342** |
| | 720 | 0.626 | 0.513 | **0.529** | **0.488** | 0.499 | 0.448 | **0.427** | **0.427** | 0.480 | 0.436 | **0.408** | **0.415** | 0.405 | 0.404 | **0.342** | **0.385** |
| Exchange | 96 | 0.084 | 0.201 | **0.079** | **0.196** | 0.106 | 0.225 | **0.100** | **0.220** | 0.087 | 0.205 | **0.082** | **0.201** | 0.094 | 0.213 | **0.088** | **0.208** |
| | 192 | 0.179 | 0.299 | **0.166** | **0.293** | 0.212 | 0.328 | **0.197** | **0.321** | 0.185 | 0.303 | **0.172** | **0.297** | 0.189 | 0.309 | **0.175** | **0.303** |
| | 336 | 0.317 | 0.403 | **0.298** | **0.394** | 0.356 | 0.433 | **0.336** | **0.424** | 0.343 | 0.420 | **0.322** | **0.411** | 0.314 | 0.403 | **0.295** | **0.394** |
| | 720 | 0.770 | 0.657 | **0.727** | **0.635** | 0.851 | 0.697 | **0.803** | **0.675** | 0.863 | 0.700 | **0.816** | **0.680** | 0.949 | 0.710 | **0.895** | **0.687** |
| Solar | 96 | 0.478 | 0.396 | **0.379** | **0.316** | 0.751 | 0.474 | **0.605** | **0.382** | 1.861 | 0.879 | **1.501** | **0.713** | 0.199 | 0.247 | **0.159** | **0.197** |
| | 192 | 0.499 | 0.393 | **0.383** | **0.314** | 0.741 | 0.496 | **0.577** | **0.403** | 2.182 | 0.990 | **1.702** | **0.807** | 0.218 | 0.267 | **0.169** | **0.216** |
| | 336 | 0.530 | 0.401 | **0.395** | **0.317** | 0.731 | 0.518 | **0.554** | **0.411** | 1.592 | 0.781 | **1.211** | **0.622** | 0.222 | 0.275 | **0.169** | **0.218** |
| | 720 | 0.552 | 0.405 | **0.417** | **0.325** | 0.730 | 0.555 | **0.558** | **0.448** | 2.053 | 0.982 | **1.570** | **0.791** | 0.220 | 0.273 | **0.167** | **0.219** |
| Weather | 96 | 0.198 | 0.231 | **0.190** | **0.226** | 0.114$^*$ | 0.141$^*$ | **0.109** | **0.138** | 0.344 | 0.270 | **0.329** | **0.264** | 0.150 | 0.196 | **0.144** | **0.192** |
| | 192 | 0.250 | 0.275 | **0.221** | **0.270** | 0.152$^*$ | 0.179$^*$ | **0.135** | **0.176** | 0.406 | 0.321 | **0.363** | **0.316** | 0.195 | 0.240 | **0.173** | **0.236** |
| | 336 | 0.324 | 0.322 | **0.263** | **0.315** | 0.219$^*$ | 0.236$^*$ | **0.180** | **0.232** | 0.407 | 0.341 | **0.334** | **0.334** | 0.255 | 0.284 | **0.211** | **0.278** |
| | 720 | 0.432 | 0.380 | **0.346** | **0.368** | 0.318$^*$ | 0.311$^*$ | **0.254** | **0.300** | 0.493 | 0.395 | **0.395** | **0.383** | 0.318 | 0.332 | **0.254** | **0.321** |
| Electricity | 96 | 0.200 | 0.273 | **0.191** | **0.268** | 0.189$^*$ | 0.270$^*$ | **0.181** | **0.265** | 0.194$^*$ | 0.257$^*$ | **0.186** | **0.253** | 0.180 | 0.271 | **0.173** | **0.267** |
| | 192 | 0.203 | 0.277 | **0.192** | **0.273** | 0.199$^*$ | 0.281$^*$ | **0.189** | **0.278** | 0.199$^*$ | 0.266$^*$ | **0.189** | **0.262** | 0.193 | 0.286 | **0.183** | **0.282** |
| | 336 | 0.222 | 0.294 | **0.208** | **0.287** | 0.225$^*$ | 0.305$^*$ | **0.212** | **0.298** | 0.215$^*$ | 0.284$^*$ | **0.202** | **0.277** | 0.213 | 0.304 | **0.200** | **0.296** |
| | 720 | 0.271 | 0.331 | **0.252** | **0.317** | 0.275$^*$ | 0.346$^*$ | **0.258** | **0.335** | 0.262$^*$ | 0.325$^*$ | **0.246** | **0.315** | 0.259 | 0.341 | **0.242** | **0.327** |
| Traffic | 96 | 0.563$^*$ | 0.319$^*$ | **0.513** | **0.301** | 0.487$^*$ | 0.275$^*$ | **0.448** | **0.260** | 0.839 | 0.335 | **0.773** | **0.318** | 0.517 | 0.344 | **0.473** | **0.325** |
| | 192 | 0.542$^*$ | 0.310$^*$ | **0.499** | **0.294** | 0.488$^*$ | 0.276$^*$ | **0.448** | **0.262** | 0.713 | 0.322 | **0.654** | **0.305** | 0.538 | 0.354 | **0.495** | **0.335** |
| | 336 | 0.558$^*$ | 0.315$^*$ | **0.512** | **0.300** | 0.503$^*$ | 0.284$^*$ | **0.462** | **0.270** | 0.698 | 0.329 | **0.642** | **0.314** | 0.571 | 0.369 | **0.525** | **0.351** |
| | 720 | 0.600$^*$ | 0.335$^*$ | **0.554** | **0.318** | 0.534$^*$ | 0.303$^*$ | **0.492** | **0.288** | 0.702 | 0.365 | **0.651** | **0.347** | 0.617 | 0.389 | **0.573** | **0.371** |

*Table 15.* Full-shot forecasting performance (MSE/MAE) of baselines and our +ChaTSFM by adopting Moirai as backbone. The lookback length is fixed to 96, and the forecasting horizons are set to $\{96, 192, 336, 720\}$. The best and second-best results are highlighted in **bold** and underline. * denotes TSFMs whose pre-training data may overlap with the evaluation datasets, leading to potential data leakage.

| Method | | Moirai$_{small}$ | | +ICM | | +PCD | | +PEFT | | +Gen-PT | | +AdaPTS | | +TTM | | +Ours | | Imp. | |
|---|---|---|---|---|---|---|---|---|---|---|---|---|---|---|---|---|---|---|---|
| | | MSE | MAE | MSE | MAE | MSE | MAE | MSE | MAE | MSE | MAE | MSE | MAE | MSE | MAE | MSE | MAE | MSE | MAE |
| ETTh1 | 96 | 0.428 | **0.402** | 0.425 | 0.404 | 0.425 | 0.404 | 0.425 | 0.404 | 0.425 | 0.404 | 0.422 | **0.402** | 0.421 | 0.403 | **0.411** | **0.402** | 3.97% | 0.19% |
| | 192 | 0.490 | 0.435 | 0.482 | 0.437 | 0.478 | 0.434 | 0.476 | 0.434 | 0.476 | 0.433 | 0.474 | 0.434 | 0.472 | 0.434 | **0.463** | **0.430** | 5.51% | 1.33% |
| | 336 | 0.549 | 0.460 | 0.545 | 0.463 | 0.545 | 0.460 | 0.543 | 0.464 | 0.541 | 0.460 | 0.538 | 0.460 | 0.539 | 0.462 | **0.525** | **0.458** | 4.37% | 0.60% |
| | 720 | 0.565 | 0.479 | 0.562 | 0.482 | 0.562 | 0.480 | 0.558 | 0.482 | 0.556 | 0.479 | 0.556 | 0.480 | 0.558 | 0.482 | **0.539** | **0.479** | 4.60% | 0.16% |
| ETTh2 | 96 | 0.335 | 0.355 | 0.334 | 0.362 | 0.333 | 0.360 | 0.331 | 0.359 | 0.330 | 0.356 | 0.328 | 0.355 | 0.327 | 0.355 | **0.326** | **0.355** | 2.68% | 0.30% |
| | 192 | 0.435 | 0.414 | 0.431 | 0.418 | 0.429 | 0.417 | 0.428 | 0.415 | 0.425 | 0.416 | 0.424 | 0.414 | 0.421 | 0.414 | **0.417** | **0.413** | 4.13% | 0.43% |
| | 336 | 0.509 | 0.463 | 0.502 | 0.466 | 0.501 | 0.464 | 0.499 | 0.464 | 0.497 | 0.463 | 0.493 | 0.463 | 0.493 | 0.463 | **0.480** | **0.459** | 5.69% | 1.03% |
| | 720 | 0.624 | 0.513 | 0.610 | 0.514 | 0.604 | 0.514 | 0.603 | 0.513 | 0.597 | 0.513 | 0.594 | 0.512 | 0.592 | 0.510 | **0.569** | **0.502** | 8.81% | 2.29% |
| ETTm1 | 96 | 0.786 | 0.521 | 0.717 | 0.510 | 0.707 | 0.506 | 0.681 | 0.499 | 0.664 | 0.492 | 0.638 | 0.488 | 0.618 | 0.481 | **0.577** | **0.472** | 26.59% | 9.54% |
| | 192 | 0.847 | 0.560 | 0.741 | 0.544 | 0.728 | 0.540 | 0.725 | 0.535 | 0.706 | 0.528 | 0.674 | 0.520 | 0.640 | 0.511 | **0.593** | **0.492** | 29.98% | 12.26% |
| | 336 | 0.936 | 0.617 | 0.810 | 0.592 | 0.796 | 0.587 | 0.784 | 0.581 | 0.762 | 0.573 | 0.723 | 0.565 | 0.666 | 0.554 | **0.616** | **0.525** | 34.18% | 15.01% |
| | 720 | 1.232 | 0.738 | 1.029 | 0.687 | 1.010 | 0.679 | 0.988 | 0.669 | 0.955 | 0.655 | 0.896 | 0.638 | 0.824 | 0.615 | **0.698** | **0.584** | 43.34% | 20.95% |
| ETTm2 | 96 | 0.212 | 0.289 | 0.211 | 0.292 | 0.210 | 0.293 | 0.209 | 0.292 | 0.209 | 0.288 | 0.208 | 0.289 | 0.209 | 0.289 | **0.207** | **0.287** | 2.35% | 0.27% |
| | 192 | 0.290 | 0.339 | 0.286 | 0.343 | 0.287 | 0.344 | 0.286 | 0.343 | 0.285 | 0.338 | 0.283 | 0.339 | 0.280 | 0.339 | **0.277** | **0.337** | 4.48% | 0.82% |
| | 336 | 0.402 | 0.403 | 0.369 | 0.409 | 0.381 | 0.407 | 0.377 | 0.408 | 0.375 | 0.402 | 0.371 | 0.401 | 0.369 | 0.401 | **0.362** | **0.396** | 9.95% | 1.93% |
| | 720 | 0.626 | 0.513 | 0.585 | 0.508 | 0.576 | 0.509 | 0.565 | 0.511 | 0.548 | 0.507 | 0.562 | 0.509 | 0.549 | 0.505 | **0.511** | **0.486** | 18.37% | 5.40% |
| Exchange | 96 | 0.084 | 0.201 | 0.082 | 0.203 | 0.082 | 0.202 | 0.081 | 0.202 | 0.080 | 0.200 | 0.079 | 0.201 | 0.079 | 0.201 | **0.078** | **0.198** | 7.14% | 1.88% |
| | 192 | 0.179 | 0.299 | 0.171 | 0.297 | 0.172 | 0.298 | 0.171 | 0.298 | 0.171 | 0.296 | 0.169 | 0.296 | 0.169 | 0.295 | **0.166** | **0.292** | 7.26% | 2.60% |
| | 336 | 0.317 | 0.403 | 0.309 | 0.401 | 0.308 | 0.401 | 0.307 | 0.400 | 0.306 | 0.398 | 0.303 | 0.396 | 0.303 | 0.399 | **0.301** | **0.395** | 5.04% | 2.17% |
| | 720 | 0.770 | 0.657 | 0.759 | 0.652 | 0.752 | 0.650 | 0.749 | 0.650 | 0.751 | 0.647 | 0.747 | 0.645 | 0.743 | 0.645 | **0.737** | **0.642** | 4.28% | 2.40% |
| Solar | 96 | 0.478 | 0.396 | 0.337 | 0.334 | 0.340 | 0.323 | 0.309 | 0.332 | 0.279 | 0.316 | 0.264 | 0.315 | 0.265 | 0.315 | **0.230** | **0.291** | 51.88% | 26.65% |
| | 192 | 0.499 | 0.393 | 0.334 | 0.331 | 0.325 | 0.329 | 0.308 | 0.333 | 0.286 | 0.322 | 0.273 | 0.301 | 0.257 | 0.312 | **0.220** | **0.284** | 55.91% | 27.87% |
| | 336 | 0.530 | 0.401 | 0.330 | 0.335 | 0.342 | 0.328 | 0.301 | 0.323 | 0.288 | 0.320 | 0.274 | 0.316 | 0.257 | 0.307 | **0.229** | **0.285** | 56.79% | 29.06% |
| | 720 | 0.552 | 0.405 | 0.336 | 0.327 | 0.325 | 0.330 | 0.307 | 0.325 | 0.295 | 0.324 | 0.274 | 0.315 | 0.268 | 0.318 | **0.233** | **0.289** | 57.78% | 28.77% |
| Weather | 96 | 0.198 | 0.231 | 0.197 | 0.230 | 0.196 | 0.230 | 0.195 | 0.229 | 0.195 | 0.228 | 0.194 | 0.228 | 0.195 | 0.228 | **0.192** | **0.227** | 3.03% | 2.06% |
| | 192 | 0.250 | 0.275 | 0.241 | 0.277 | 0.240 | 0.276 | 0.239 | 0.273 | 0.233 | 0.274 | 0.234 | 0.274 | 0.229 | 0.274 | **0.224** | **0.274** | 10.39% | 0.65% |
| | 336 | 0.324 | 0.322 | 0.303 | 0.320 | 0.303 | 0.319 | 0.297 | 0.324 | 0.290 | 0.319 | 0.286 | 0.318 | 0.291 | 0.318 | **0.272** | **0.319** | 16.04% | 1.17% |
| | 720 | 0.432 | 0.380 | 0.401 | 0.380 | 0.397 | 0.381 | 0.390 | 0.378 | 0.378 | 0.375 | 0.362 | 0.375 | 0.365 | 0.374 | **0.345** | **0.372** | 20.13% | 2.31% |
| Electricity | 96 | 0.200 | 0.273 | 0.195 | 0.271 | 0.192 | 0.274 | 0.192 | 0.273 | 0.189 | 0.270 | 0.186 | 0.269 | 0.189 | 0.269 | **0.183** | **0.268** | 8.50% | 2.11% |
| | 192 | 0.203 | 0.277 | 0.194 | 0.279 | 0.193 | 0.277 | 0.194 | 0.276 | 0.189 | 0.276 | 0.192 | 0.275 | 0.188 | 0.276 | **0.184** | **0.273** | 9.35% | 1.72% |
| | 336 | 0.222 | 0.294 | 0.213 | 0.294 | 0.210 | 0.295 | 0.210 | 0.293 | 0.207 | 0.292 | 0.206 | 0.289 | 0.203 | 0.290 | **0.196** | **0.286** | 11.71% | 2.98% |
| | 720 | 0.271 | 0.331 | 0.253 | 0.328 | 0.253 | 0.328 | 0.250 | 0.322 | 0.245 | 0.319 | 0.240 | 0.316 | 0.243 | 0.319 | **0.233** | **0.309** | 14.02% | 6.86% |
| Traffic | 96 | 0.563* | 0.319* | 0.537 | 0.308 | 0.536 | 0.306 | 0.533 | 0.313 | 0.532 | 0.308 | 0.525 | 0.308 | 0.521 | 0.308 | **0.511** | **0.301** | 9.23% | 5.87% |
| | 192 | 0.542* | 0.310* | 0.523 | 0.304 | 0.520 | 0.303 | 0.518 | 0.303 | 0.512 | 0.300 | 0.508 | 0.298 | 0.512 | 0.296 | **0.495** | **0.293** | 8.67% | 5.72% |
| | 336 | 0.558* | 0.315* | 0.528 | 0.308 | 0.522 | 0.307 | 0.524 | 0.305 | 0.516 | 0.304 | 0.519 | 0.302 | 0.513 | 0.302 | **0.507** | **0.298** | 9.13% | 5.63% |
| | 720 | 0.600* | 0.335* | 0.579 | 0.331 | 0.578 | 0.326 | 0.577 | 0.325 | 0.573 | 0.326 | 0.568 | 0.324 | 0.567 | 0.320 | **0.554** | **0.318** | 7.66% | 5.29% |

