# OpenReview forum: "Channel Adapter for Time Series Foundation Models in Zero-Shot Multivariate Forecasting"
_ICML.cc/2026/Conference — ICML 2026 regular_

### Official Review · Reviewer_dr8k · 2026-02-27

**Soundness:** 2
**Presentation:** 2
**Significance:** 2
**Originality:** 2
**Overall Recommendation:** 3
**Confidence:** 5

**Summary:**

This paper propose a lightweight plugin for spatial information interaction on the time series foundation models， thereby further improving prediction accuracy.

**Compliance With Llm Reviewing Policy:**

Affirmed.

**Final Justification:**

Based on the author's current response, I have decided to raise my rating from 2 to 3. The reason is that this paper requires significant revisions to clarify its viewpoints and process before it can be published; therefore, I am inclined to decline its acceptance.

**Key Questions For Authors:**

See weaknesses

**Limitations:**

yes

**Strengths And Weaknesses:**

Strengths：

This is a very interesting research direction. Due to the heterogeneity of data from different domains during the pre-training stage, many works perform channel-independent modeling when building the time series foundation models. However, the spatiotemporal correlation of multi-channel data is not considered in the downstream task stage, which will affect the model performance.

Weaknesses：

1、	In the abstract, this paper mentions that most time series foundation models perform channel-independent modeling, resulting in poor spatiotemporal information mining capabilities. However, the foundation model MOIRAI used in the experimental verification in this paper is not channel-independent modeling. It is recommended that the authors re-examine the correspondence between the abstract and main text of the paper, otherwise there will be contradictions.

2、	The data processing flow in this paper is unclear. The purpose of the first step, Generative Domain Selection, mentioned in the overall framework, which selects the top-k domain data from the pre-training dataset. It is also unclear what the relationship is between this step and the second step, Geometric Metric Adaptation. It is recommended that the authors outline a clear data flow to demonstrate the relationship between them.

3、	During the Generative Domain Selection phase, was the pre-training dataset reconstructed by the authors themselves, or was it the pre-training dataset corresponding to the foundation model used?

---

> ### Author Rebuttal · Authors · 2026-03-30
>
> ***Thank you very much for your valuable and constructive comments. We have carefully responded to your concerns and hope that our revisions will help alleviate them.***
>
> **Q1**: Consistency between the abstract CI claim and Moirai.
>
> **Reply-1**: We thank the reviewer for this precise observation. We would like to clarify that the Moirai used in our paper is Moirai 2.0 (moirai-2.0-R-small in Table 10), not Moirai 1.0. Moirai 1.0 introduced Any-variate Attention to support multivariate interaction, whereas Moirai 2.0 explicitly dropped support for multivariate forecasting, as acknowledged by its authors: “Seeing minimal benefit in doing so, we have dropped support for multivariate forecasting and the use of covariates in Moirai 2.0.” Thus, the use of Moirai 2.0 does not contradict the CI setting described in our abstract.
> Following your suggestion, we will revise the manuscript to refer to Moirai 2.0 consistently throughout, clarify this distinction explicitly in the abstract, main text, and experimental setup, and add the corresponding reference [Moirai 2.0: When Less Is More for Time Series Forecasting, Liu et al., 2025.]
>
>
> **Q2**: Unclear data flow between Generative Domain Selection and Geometric Metric Adaptation.
>
> **Reply-2**: We thank the reviewer for this valuable suggestion. We agree that clarifying the data flow improves the paper’s comprehensibility, and we address each point below.
>
> - **Regarding the purpose of Generative Domain Selection**: This step operates before pre-training and serves as a data curation strategy. Given a large candidate pool of multivariate datasets (1.2T observations from LOTSA, ECG, ERA5-Land, and Time-300B), naively using all available data introduces redundancy and noise, while random sampling risks missing rare but critical inter-channel dependency patterns. Our greedy selection criterion (Eq. 1) therefore constructs a compact yet maximally diverse pre-training corpus. The output of this step is the selected 10% subset, which serves as the training corpus for the Chada plugin, most critically for training the Geometric Hyper-Network $H_\theta$.
>
> - **Regarding the relationship between two steps**: The two steps operate at distinct stages but are directly connected. Generative Domain Selection (Step 1, offline) determines the domain subset on which $H_\theta$ is trained, while Geometric Metric Adaptation (Step 2, online inference) applies the learned mapping to generate domain-adaptive metric parameters from the statistical features of a new target dataset. Importantly, the selected top-k domains are used to construct the pre-training corpus, rather than being retrieved again during zero-shot inference. Without a sufficiently diverse corpus from Step 1, $H_\theta$ would overfit to narrow domain patterns and generalize poorly to unseen domains.
>
> **Action-2**: To resolve this for readers, we will add the following Data Flow Overview at the beginning of Section 4: Chada operates in two phases. In the offline corpus curation phase (Sec. 4.1), Generative Domain Selection constructs a geometrically diverse pre-training corpus for learning $H_\theta$.  In the online inference phase (Secs. 4.2–4.3), $H_\theta$ takes the statistical features $\phi(X)$ of a new target dataset as input, generates domain-adaptive metric parameters on-the-fly, and supports the subsequent Channel-Adapter Dependency Modeling.
>
>
> **Q3**: Was the pre-training corpus author-constructed or backbone-provided?
>
> **Reply-3**: We thank the reviewer for this important clarifying question. The pre-training corpus used for the Generative Domain Selection was independently constructed by us, rather than adopting the original pre-training datasets of any backbone TSFM. Concretely, we assembled a 1.2T-observation candidate pool from diverse public collections and applied Generative Domain Selection (Eq. 1) to extract a compact 10% subset that maximizes geometric diversity in inter-channel dependency patterns. Although some public sources may overlap with data used by existing TSFMs, our corpus construction and selection process are entirely independent.
> This curated corpus is used to train the Chada, while TSFM parameters remain frozen. In other words, we do not reconstruct the backbone model itself. The same independently curated corpus is used to train Chada across all evaluated frozen TSFMs, rather than constructing a separate corpus for each backbone. This keeps Chada’s training data pipeline independent of any specific backbone while preserving the frozen-backbone setting.
>
> **Action-3**: We will revise Section 5 to explicitly describe the independent curation of Chada’s training corpus and its strict separation from the pre-training data of any backbone TSFM.
>
> ***We sincerely hope our detailed responses have addressed your concerns and clarified the strengths of our work. If you find our revisions satisfactory, we would greatly appreciate it if you could consider raising your score and supporting our paper.***

---

> > ### Author Rebuttal · Reviewer_dr8k · 2026-04-03
> >
> > Thank you for the author's reply; I will increase my rating accordingly.

---

> > > ### Author Response · Authors · 2026-04-05
> > >
> > > Dear Reviewer dr8k,
> > >
> > > Thank you very much for your thoughtful and constructive comments. We are especially grateful for your recognition of the importance of this research direction, the practical motivation behind our work, and the value of introducing a lightweight plugin to better capture spatiotemporal correlations in time series foundation models.
> > >
> > > In the revised paper, we will explicitly refer to Moirai 2.0 throughout, add a brief data flow description in Section 4, and clarify the construction and role of our pre-training corpus. If you have any additional questions or concerns about our work, we would be very happy to provide further clarification. We sincerely thank you again for your valuable suggestions, which have helped us improve the quality and clarity of this paper.
> > >
> > > Best regards,
> > >
> > > Authors of Chada

---

### Official Review · Reviewer_VBrp · 2026-03-07

**Soundness:** 4
**Presentation:** 4
**Significance:** 3
**Originality:** 4
**Overall Recommendation:** 5
**Confidence:** 5

**Summary:**

This paper addresses an interesting and important problem: most existing Time Series Foundation Models adopt a channel-independent pre-training paradigm, which makes it difficult to use the critical cross-channel dependencies in multivariate systems. To address this limitation, the authors propose Chada, a lightweight, plug-and-play channel adapter that supports zero-shot adaptation and injects cross-variable interaction information into frozen TSFMs. Chada contains three main components: Generative Domain Selection for constructing an informative and diverse pre-training subset, Geometric Metric Adaptation for mitigating cross-domain similarity distortion via a Hyper-Network, and Channel-Partial Modeling for selectively aggregating useful cross-channel information through sparse topology induction and gated residual refinement. The method is evaluated on four representative TSFM backbones and nine multivariate forecasting benchmarks. The results show that Chada consistently improves forecasting performance while adding only minimal memory and computational overhead.

**Compliance With Llm Reviewing Policy:**

Affirmed.

**Final Justification:**

The authors’ rebuttal effectively addressed my core concerns: they clarified the motivation for domain-adaptive metric calibration, distinguished formal analysis from conceptual intuition, and explicitly defined Chada’s scope as a lightweight post-hoc plugin.  I maintain my recommendation of Accept (5).

**Key Questions For Authors:**

1. Authors should clarify more explicitly why domain-adaptive metric calibration is the most appropriate design choice for cross-domain zero-shot multivariate forecasting? The paper provides reasonable intuition and empirical support, but a more direct explanation of why this formulation is preferable to using a single shared similarity metric or other simpler cross-channel interaction schemes would strengthen the presentation.
2. Which parts of the theoretical analysis are intended as formal support for the method design, and which parts are better interpreted as conceptual motivation or modeling intuition? Authors should clarify these points to help readers better understand the paper and assess its contribution more accurately.
3. Authors should discuss more clearly the intended scope and boundary of Chada as a lightweight post-hoc plugin. As presented, Chada mainly performs output-level cross-channel refinement rather than deeply modifying the backbone’s internal spatiotemporal representation learning. I do not view this as a flaw; in fact, this is part of the practical appeal of the method. However, a clearer discussion of when this plugin-style intervention is particularly suitable, and what limitations remain compared with deeper backbone-level integration, would strengthen the paper.
4. Authors should explain more explicitly why Channel-Partial modeling should be viewed as a principled design choice rather than primarily a computational simplification. The paper argues that sparse, selective aggregation is preferable to denser cross-channel interaction because it helps avoid spurious correlations while preserving the backbone’s temporal priors. This is a compelling intuition, but a more explicit explanation would make the overall method narrative stronger.
5. Although the paper is overall well written, there appear to be a few typos in the manuscript, for example around Equation 12. Addressing these minor issues would further improve the presentation quality.

**Limitations:**

Yes

**Strengths And Weaknesses:**

Strengths:
1. The paper addresses an important problem: current time series foundation models adopt channel-independent pre-training paradigm, making them difficult to capture the cross-channel dependencies that are essential in multivariate systems. To address this, the paper proposes Chada, an innovative plug-and-play channel adapter that injects multivariate interaction information into frozen TSFMs and improves forecasting performance while preserving efficiency and deployment scalability.
2. The method is well-motivated and supported by both theoretical intuition and empirical evidence. The paper presents a coherent design pipeline with each component being accompanied by corresponding motivation and experimental validation. In particular, the discussion on correcting cross-domain geometric distortion is insightful and may inspire future work beyond this specific method.
3. Chada demonstrates strong generality across model families. A notable strength of the paper is that Chada is evaluated on four representative TSFM backbones spanning different architectural paradigms, including MLP-based, decoder-only, encoder-only, and encoder-decoder models. The fact that the method consistently improves performance across such diverse backbones provides strong evidence that the proposed plugin captures a broadly useful modeling principle rather than being tailored to a single architecture.
4. Experiments are convincing. This work includes comprehensive experiments on 4 TSFM backbones and 9 benchmarks along with additional analyses such as efficiency evaluation, domain representation analysis, metric adaptation analysis, and multiple case studies. This broad empirical coverage substantially strengthens the paper and gives readers confidence that the observed gains are robust and well supported.

Weaknesses:
1. Some key design choices would benefit from clearer motivation and positioning. In particular, the paper could more clearly justify the role of domain-adaptive metric calibration and Channel-Partial modeling in the overall framework.
2. The relationship between the theoretical analysis and the final implemented method could be clarified further. A clearer distinction between formal justification and conceptual motivation would improve the paper’s technical transparency.
3. The paper would benefit from a clearer discussion of the method’s scope and a more careful polishing of the presentation. This includes the intended boundary of Chada as a lightweight post-hoc plugin, as well as a few minor typos and presentation issues.

---

> ### Author Rebuttal · Authors · 2026-03-30
>
> ***We thank the reviewer for recognizing our plug-and-play innovation, theoretical and empirical solidness, cross-TSFM generality, and comprehensive evaluations.***
>
> **W1&Q1**: Authors should clarify more explicitly why domain-adaptive metric calibration is appropriate design.
>
> **Reply-1**: We sincerely thank the reviewer for this insightful suggestion. Domain-adaptive calibration is a theoretical necessity for zero-shot forecasting, justified by two factors:
> - *Preventing Geometric Distortion*: A shared metric wrongly assumes a homogeneous geometric space across datasets. As per our SDE formulation (Def 4.1) and rigorously proved in Theorem 4.3, applying a fixed metric across domains with varying diffusion dynamics mathematically guarantees geometric distortion. By dynamically calibrating the metric space via statistical fingerprints, Chada aligns embedding distances with true dependency strengths. This is validated by the severe degradation of the Learnable Global baseline.
> - *Mitigating Spurious Correlations*: Simpler schemes introduce structural noise. While full-shot fine-tuning can gradually unlearn this, zero-shot models cannot. Our calibration dynamically induces a sparse, high-fidelity topology before message passing, ensuring inter-channel context provides genuine information gain.
>
> **Action-1**: We will add a dedicated discussion in Section 4.2 to highlight the necessity of our design.
>
> **W2&Q2**: Clarify which parts of the theoretical analysis provide formal support versus conceptual motivation.
>
> **Reply-2**: We sincerely thank the reviewer for pointing this out. We clarify our theoretical boundaries as follows:
>
> - *Conceptual Motivation*: Def 4.1 and Prop 4.2 use SDEs as a conceptual lens. They provide physically grounded intuition for summarizing domains via low-order trajectory moments, motivating our feature extraction framework rather than providing formal learning bounds.
> - *Formal Support*: (1). Theorem 4.3 rigorously proves a lower-bounded error when applying fixed metrics across heterogeneous domains, serving as the strict mathematical justification for the necessity of our Geometric Hyper-Network. (2). Coverage Guarantee Theorem formally proves that our greedy domain selection achieves an additive $k$-center bound, mathematically guaranteeing a robust and diverse pre-training subset under budget constraints.
>
> **Action-2**: To ensure clarity, we will add a brief Theoretical Roadmap at the beginning of Section 4 to map these boundaries for readers.
>
> **W3&Q3**: Clarify the intended scope, suitability, and limitations of Chada as a post-hoc plugin.
>
> **Reply-3**: We thank the reviewer for recognizing Chada's practical appeal. We clarify its scope and boundaries:
>
> - *Suitability*: Chada is ideal for: (1) Black-box models, serving as the only viable approach when internal representations are inaccessible; (2) Preserving priors, avoiding the catastrophic forgetting inherent in deep fine-tuning; and (3) Resource-constrained deployment, bypassing the layer-wise quadratic complexity of deep cross-attention.
> - *Limitations*: Operating post-hoc means Chada relies on the backbone's initial univariate feature extraction and cannot retroactively correct early-stage temporal failures. While deep integration enables joint reasoning where spatial context guides temporal extraction layer-by-layer, Chada trades this joint optimization for architectural agnosticism and extreme efficiency.
>
> **Action-3**: We will incorporate this discussion into the Limitations section of the camera-ready version.
>
> **Q4**: Explain why Channel-Partial modeling is a principled design choice.
>
> **Reply-4**: We thank the reviewer. CP modeling is fundamentally a principled inductive bias, justified by:
>
> - *Physical Causality*: Real-world systems possess inherently sparse causal structures. Dense interactions force models to fit structural noise by learning near-zero weights for irrelevant channels. CP natively mirrors true physical sparsity.
> - *Zero-Shot Necessity*: While fine-tuning might unlearn spurious correlations, zero-shot settings lack task-specific gradients to suppress noise. Thus, sparse Top-K aggregation is a crucial topological regularizer protecting frozen priors.
>
> Table 5 confirms this: Full Attention strictly underperforms CP. If CP were merely a computational simplification, dense attention would theoretically bound its performance. Instead, it degrades accuracy via noise injection.
>
> **Action-4**: We will explicitly frame CP as a topological regularizer mirroring sparse causality in the revision of Section 4.3.
>
> **Q5**: Address minor typos.
>
> **Reply-5**: We thank the reviewer for their careful reading. We have corrected the formatting duplication around Equation 12 and will thoroughly proofread the entire manuscript for the camera-ready version.
>
> ***If our responses resolve your concerns, we respectfully hope you will maintain your positive rating to support our work.***

---

> > ### Author Rebuttal · Reviewer_VBrp · 2026-04-02
> >
> > The authors have provided comprehensive, point-by-point responses to all my original concerns. I have no further questions and have decided to maintain my score.

---

> > > ### Author Response · Authors · 2026-04-02
> > >
> > > Dear Reviewer VBrp
> > >
> > > Thank you so much for your encouraging feedback and for your support toward the acceptance of our paper. We sincerely appreciate your time and constructive comments throughout the review process.
> > >
> > > Best regards,
> > >
> > > Authors of Chada

---

### Official Review · Reviewer_7Mon · 2026-03-09

**Soundness:** 3
**Presentation:** 3
**Significance:** 3
**Originality:** 3
**Overall Recommendation:** 5
**Confidence:** 5

**Summary:**

This paper proposes Chada, a lightweight plug-and-play channel adapter that enhances the zero-shot forecasting performance of time series foundation models by recovering multivariate correlations. It presents a greedy selection strategy to obtain pre-training dataset with diverse statistical properties under limited scale. It then introduces a Hyper-Network that dynamically generates domain-specific metrics to construct the correlation matrix between channels. The channel outputs aggregated according to the correlations and the outputs of time series forecasting backbone are fused via a gating mechanism to obtain the final results. Comprehensive experiments on numerous datasets and settings consistently verify the superiority of Chada. Further experimental analyses validate the effectiveness and rationality of components.

**Compliance With Llm Reviewing Policy:**

Affirmed.

**Final Justification:**

The authors clearly present their contributions and innovations, and effectively address our concerns during the rebuttal. We recommend accept and have raised our score accordingly.

**Key Questions For Authors:**

Please see the weaknesses.

**Limitations:**

yes

**Strengths And Weaknesses:**

Strengths:

1. It proposes a novel perspective on Domain Selection, which ensure the diversity of pre-trained dataset and effectively support downstream tasks under resource-constrained conditions. This provides insights for dataset construction in relevant time series tasks.
2. The Hyper-Network that dynamically generates the domain-adaptive metric can effectively measure the multivariate correlations in two different scenarios: high-diffusion domains (e.g., financial markets) and low-diffusion domains (e.g., power grids).
3. The structure and context of the paper are clear, with rich and comprehensive experiments.

Weaknesses:

1. Adaptive-AvgPool in Eq.(3) is used to preserve the temporal patterns. However, important temporal patterns may still exist in each temporal bin, which can be damaged by the pooling operation. In addition, the selection of the key hyperparameter $S$ (temporal bins) lacks further discussion. As a fixed value of $S=16$ is adopted in Chada, it is unclear how it impacts datasets with different granularities.
2. While the parameters of Time Series Foundation Model (TSFM) is frozen, its outputs are used as the sole input to Chada (Figure 1). It means that the errors generated in TSFM will be directly passed to Chada, resulting in spurious correlations modeling. In zero-shot setting, such errors will be further accumulated and amplified, thereby degrading the prediction performance.

---

> ### Author Rebuttal · Authors · 2026-03-30
>
> ***We sincerely thank you for the time and effort dedicated to reviewing our paper. We are highly appreciate your recognition of the innovation of our work and the comprehensiveness of our experiments.***
>
> **Q1**: Eq.(3) may lose important intra-bin temporal patterns, and the choice of S=16 is insufficiently justified.
>
> **Reply-1**: We thank the reviewer for this helpful comment and address the two concerns separately.
>
> - **On the use of Adaptive-AvgPool**: Adaptive-AvgPool is not designed to preserve all fine-grained intra-bin temporal patterns. Rather, its purpose is to construct a horizon-agnostic channel embedding for inter-channel dependency modeling, while the frozen TSFMs remain responsible for modeling fine-grained intra-channel temporal dynamics. Therefore, Eq.(3) serves the channel encoder branch for graph construction, rather than modifying the backbone’s original temporal modeling pathway. Under this decoupled design, macroscopic trajectory statistics act as the primary signal for this branch. Preserving all intra-bin variations would introduce unnecessary high-frequency noise.
>
> - **On the choice of S**: S controls the resolution of the horizon-agnostic channel embedding: a smaller S over-compresses the predicted trajectory, whereas a larger S makes the induced channel geometry overly sensitive to local temporal variations. To examine this sensitivity, we conduct an ablation study with $S \in$ \{4, 8, 16, 32\}. We select datasets spanning an extreme spectrum of sampling granularities and distinct physical domains. In Table 1, forecasting performance is stable across all tested values of S. While specific datasets might see marginal gains at S=8, S=16 consistently yields optimal or near-optimal performance. This confirms S=16 as a universally robust empirical default that spares the need for dataset-specific tuning.
>
> **Table 1: Ablation study of S using Moirai (MSE / MAE)**.
> |Data|S=4|S=8|S=16 (Default)|S=32|
> |-|-|-|-|-|
> |Weather (10-min)|0.256/0.295|0.256/0.295|0.255/0.295|0.256/0.296|
> |ETTm2 (15-min)|0.344/0.377|0.344/0.376|0.343/0.376|0.346/0.378|
> |ETTh1 (Hourly)|0.498/0.441|0.496/0.440|0.496/0.440|0.498/0.442|
> |Exchange (Daily)|0.319/0.381|0.317/0.378|0.318/0.380|0.318/0.381|
> |
>
> **Action-1**: We will revise Section 4.2 to explicitly clarify the decoupled role of Adaptive-AvgPool and include this ablation analysis of S.
>
> **Q2**. Chada appears to use frozen TSFM outputs as its sole input, which may propagate TSFM prediction errors, induce spurious inter-channel correlations, and amplify errors in the zero-shot setting.
>
> **Reply-2**: We thank the reviewer for raising this important concern. We would like to first clarify a misunderstanding regarding the input structure of Chada, and then explain how our design effectively addresses error propagation.
>
> - **Dual-pathway design**:  While the channel embeddings are derived from the TSFM prediction, the domain-adaptive metric parameters [γ, β] in Eq.(4) are generated by the Geometric Hyper-Network $H_\theta(\phi(X))$. Crucially, $\phi(X)$ is computed directly from the original historical input, entirely independent of the TSFM's predictions. This is our architectural safeguard: even if predicted outputs carry errors, the metric space used to measure inter-channel similarity is calibrated by reliable historical observations. In Eq.(5), the potentially noisy embeddings are projected into this calibrated metric space, where γ adaptively down-weights noise-dominated dimensions in high-diffusion domains. Thus, spurious correlations driven by TSFM prediction errors are mitigated rather than directly propagated.
>
> - **Gated residual mechanism**: The final prediction is controlled by a learned confidence gate $g \in (0,1)$, which adaptively modulates the correction intensity based on whether the inter-channel context provides reliable information gain. The output weight of $\delta$ is initialized to zero and the gate bias to -2.5, ensuring $\hat{X}^{predict} \approx \hat{X}^{model}$ at the start of training. This ensures Chada only injects inter-channel corrections when they provide genuine information gain, and even in the presence of residual noise, the gating mechanism prevents the final output from being worse than the original TSFM prediction.
>
> - **Empirical Validation**: Chada improves frozen TSFMs in zero-shot settings, with large gains on strongly coupled data (22.4% MSE reduction on Solar and 7.9% on Traffic). These results suggest that Chada can mitigate noisy or spurious inter-channel dependencies.
>
> **Action-2**: We acknowledge that the dual-pathway nature of Chada was not sufficiently highlighted. We will revise Section 4.2 to explicitly emphasize that $\phi(X)$ serves as a structural calibrator against TSFM prediction errors.
>
> ***We sincerely hope our detailed explanations have addressed your concerns. If you find our revisions satisfactory, we would greatly appreciate it if you could consider raising your score and supporting our paper.***

---

> > ### Author Rebuttal · Reviewer_7Mon · 2026-04-02
> >
> > Thank the authors for their detailed rebuttal. Our concerns have been addressed, and we maintain our positive score.

---

> > > ### Author Response · Authors · 2026-04-02
> > >
> > > Dear Reviewer 7Mon,
> > >
> > > Thank you so much for your encouraging feedback. Following your valuable suggestions, we will incorporate into the revised paper the clarification of the decoupled role of Adaptive-AvgPool, the ablation analysis on the choice of S, and the explanation of Chada’s dual-pathway design and error-mitigation mechanism.
> > >
> > > We sincerely appreciate your time and constructive comments throughout the review process.
> > >
> > > Best regards,
> > >
> > > Authors of Chada

---

### Official Review · Reviewer_Rr57 · 2026-03-13

**Soundness:** 3
**Presentation:** 3
**Significance:** 3
**Originality:** 3
**Overall Recommendation:** 4
**Confidence:** 4

**Summary:**

The paper proposes Chada, a plug-and-play channel adapter that enhances zero-shot multivariate forecasting for pre-trained time-series foundation models. Chada comprises three components: generative domain selection to curate a compact yet diverse pretraining set of multivariate dependencies; a geometric hyper-network to rectify inter-domain similarity distortions; and a channel-partial sparse dependency module to inject cross-channel information without disturbing learned temporal priors. Experiments across four backbones show consistent zero-shot gains with minimal compute/parameter overhead.

**Compliance With Llm Reviewing Policy:**

Affirmed.

**Final Justification:**

I am satisfied with the author’s rebuttal and am willing to maintain my initial positive assessment.

**Key Questions For Authors:**

Please refer to Weaknesses.

**Limitations:**

Yes

**Strengths And Weaknesses:**

## **Strength**

* This paper is well-organized and easy to follow.

* Using a plug-and-play manner to improve TSFMs' performance and shift traditional univariate modeling to multivariate modeling is promising and makes sense beyond using large-scale fine-tuning to achieve it.

* Comprehensive experiments demonstrate its effectiveness.

---

## **Weaknessne**

* It is unclear whether a single Chada is trained and deployed across different backbones or whether one plugin is trained per backbone. My primary concern is the cross-backbone transfer, which is an important test of TSFM architecture-agnostic claims. It would be better to make a clearer statement.

* This paper could better contextualize against recent zero-shot multivariate adapters or plugins or strong zero-shot multivariate-capable TSFMs, such as encoder-only models with channel attention trained for multivariate time series datasets.

* Lack of some key hyperparameter sensitivity studies, such as embedding dimension $D$, pooling bins $S$, neighborhood size distribution $K_D$, $\lambda$, and ε in selection.

---

> ### Author Rebuttal · Authors · 2026-03-30
>
> ***We thank the reviewer for the thoughtful review. We are encouraged that you find Chada promising and well supported by our comprehensive experiments.***
>
> **Q1**: Clarification on whether Chada is trained for all backbones or one per backbone, and cross-backbone transfer.
>
> **Reply-1**. We thank the reviewer for raising this important point. One dedicated Chada is trained per backbone for optimal performance. Our architecture-agnostic claim refers to plug-and-play integration across heterogeneous backbones, rather than sharing one identical weights. Unlike internal adaptation methods like LoRA, which must be customized for architecture-specific modules, Chada operates purely on the output side. As long as a backbone provides the historical sequence and final prediction, Chada can be attached without modifying any internal components.
>
> - **Why dedicated training is still beneficial**: Chada is designed with two functionally distinct components. The Geometric Hyper-Network performs domain-adaptive metric calibration from input statistics, which is intended to capture a transferable geometric prior for inter-channel dependency estimation. In contrast, the gated residual refinement operates on the host backbone’s prediction and thus must adapt to the prediction characteristics of that specific backbone. Therefore, plug-in compatibility is architecture-agnostic, while optimal parameterization remains backbone-specific.
>
> - **Cross-Backbone Experiment**: We conduct a transfer experiment between Moirai and Chronos. In Table 1, transferred plugins outperform frozen backbones, showing that metric adaptation generalizes across architectures. Meanwhile, dedicated plugins perform best, confirming that the refinement stage requires host-specific adaptation.
>
> **Table 1: Cross-backbone transfer performance (MSE / MAE) in zero-shot setting.**
> |Backbone|ETTm2|Solar|Exchange|
> |-|-|-|-|
> |Moirai|0.383/0.386|0.515/0.399|0.338/0.390|
> |+Chada (Trained on Chronos)|0.360/0.381|0.428/0.364|0.331/0.384|
> |+Chada (Trained on Moirai)|0.343/0.376|0.394/0.318|0.318/0.380|
> | Chronos|0.351/0.366|1.922/0.908|0.369/0.407|
> | +Chada (Trained on Moirai)|0.345/0.360|1.810/0.825|0.358/0.403|
> | +Chada (Trained on Chronos)|0.320/0.357|1.496/0.733|0.348/0.397|
> |
>
> **Action-1**: We will explicitly clarify this distinction in Section 4 and add the cross-backbone transfer analysis to the appendix to better support the meaning of the architecture-agnostic claim.
>
>
> **Q2**: Lack of contextualization against recent zero-shot multivariate adapters, plugins, and native multivariate-capable TSFMs
>
> **Reply-2**: We thank the reviewer for this insightful feedback. We position these advances along our Table 1 capability axes:
>
> - **Comparison with Native Multivariate TSFMs**: Recent models like GTT (Feng et al., 2024) and Chronos-2 (Ansari et al., 2025) jointly train spatio-temporal dynamics from scratch. However, as noted in Chronos-2, "jointly modeling multiple variates in a zero-shot setting remains an open challenge" due to spurious correlations. Instead, Chada explicitly decouples temporal and spatial induction. Relying on the robust temporal priors of CI backbones, it handles cross-channel dependencies as a domain-adaptive topology. Thus, Chada is not a replacement for native multivariate TSFMs, but a lightweight zero-shot enhancement layer for established CI models.
>
> - **Comparison with Recent Zero-Shot Adapters**: Following Table 1, recent adapters differ from Chada in two ways: (i) Parameter-updating adapters achieve multivariate adaptation through target-domain tuning. While effective, they trade strict zero-shot efficiency for performance by requiring gradient updates. (ii) Internal zero-shot plugins inject cross-channel modules inside the transformer blocks. This internal modification requires careful tuning to avoid interfering with the temporal processing of the frozen backbone. Chada is a strictly zero-shot, output-side plugin. It infers domain-adaptive dependencies from historical inputs to refine final predictions. This non-invasively augments CI models, requiring zero internal changes or gradient updates.
>
> **Action-2**: We will expand Section 2 and update Table 1 and recent internal adapters. We will incorporate the design-space analysis above to objectively contextualize Chada's architectural choices.
>
>
> **Q3**: Lack of hyperparameter sensitivity studies
>
> **Reply-3**: We thank the reviewer for pointing out this omission. While Table 9 lists default hyperparameters, we will add sensitivity experiments to the Appendix (results: https://anonymous.4open.science/r/ICML2026-1AF5/hyperparam_sensitivity.png). The figure demonstrates highly stable performance. $K_{D}$ is excluded from this static search as it is dynamically generated by the Hyper-Network.
>
> ***We sincerely hope that our explanation helps alleviate your concerns and provides a clearer understanding of our work. Could you kindly champion our paper if you think all concerns are addressed ?***

---

> > ### Author Rebuttal · Reviewer_Rr57 · 2026-04-02
> >
> > I appreciate the authors' detailed rebuttals. I am generally satisfied with these responses and stand by my positive score.

---

> > > ### Author Response · Authors · 2026-04-02
> > >
> > > Dear Reviewer Rr57,
> > >
> > > Thank you so much for your encouraging feedback and for your support toward the acceptance of our paper.
> > >
> > > We are grateful that you found our rebuttal helpful and that your concerns have been adequately addressed. Following your suggestions, we will incorporate the clarification on the backbone-specific training setting, the discussion on recent zero-shot multivariate adapters, as well as the additional hyperparameter sensitivity analysis into the revised paper.
> > >
> > > We sincerely appreciate your time and constructive comments throughout the review process.
> > >
> > > Best regards,
> > >
> > > Authors of Chada

---

### Decision · Program_Chairs · 2026-04-30

**Decision:**

Accept (regular)

**Comment:**

Three reviewers support acceptance. The authors’ rebuttal thoroughly addressed the negative reviewer’s main concerns. Although that reviewer still asks for major clarifications, the requested changes are limited (abstract consistency, data flow, dataset description) and can be fixed in the final revision. The AC therefore recommends accept, with the requirement that the authors make the following revisions in the camera‑ready version:

1、Clearly state the use of Moirai 2.0 in the abstract and text, and explain its consistency with the channel‑independent claim.

2、Add a clear data‑flow overview (a diagram is fine) showing the offline curation phase, online inference phase, and the relationship between Generative Domain Selection and Geometric Metric Adaptation.

3、Explicitly describe how the pre‑training corpus was independently built, including its size, sources, and separation from any backbone model’s pre‑training data.

The authors have already agreed to these changes, so the final manuscript should meet the standard. Overall, this paper is valuable to the community, but it still requires careful revisions in the final version.